# Aurora A depletion reveals centrosome-independent polarization mechanism in *Caenorhabditis elegans*

Kerstin Klinkert[1], Nicolas Levernier[2,3], Peter Gross[4], Christian Gentili[1], Lukas von Tobel[1], Marie Pierron[1], Coralie Busso[1], Sarah Herrman[1], Stephan W Grill[4,5,6], Karsten Kruse[2,3,7], Pierre Gönczy[1]*

[1]Swiss Institute for Experimental Cancer Research (ISREC), School of Life Sciences, Swiss Federal Institute of Technology Lausanne (EPFL), Lausanne, Switzerland; [2]Department of Biochemistry, University of Geneva, Geneva, Switzerland; [3]Department of Theoretical Physics, University of Geneva, Geneva, Switzerland; [4]BIOTEC, TU Dresden, Dresden, Germany; [5]Max Planck Institute of Molecular Cell Biology and Genetics, Dresden, Germany; [6]Cluster of Excellence Physics of Life, TU Dresden, Dresden, Germany; [7]National Center of Competence in Research Chemical Biology, University of Geneva, Geneva, Switzerland

**Abstract** How living systems break symmetry in an organized manner is a fundamental question in biology. In wild-type *Caenorhabditis elegans* zygotes, symmetry breaking during anterior-posterior axis specification is guided by centrosomes, resulting in anterior-directed cortical flows and a single posterior PAR-2 domain. We uncover that *C. elegans* zygotes depleted of the Aurora A kinase AIR-1 or lacking centrosomes entirely usually establish two posterior PAR-2 domains, one at each pole. We demonstrate that AIR-1 prevents symmetry breaking early in the cell cycle, whereas centrosomal AIR-1 instructs polarity initiation thereafter. Using triangular microfabricated chambers, we establish that bipolarity of *air-1(RNAi)* embryos occurs effectively in a cell-shape and curvature-dependent manner. Furthermore, we develop an integrated physical description of symmetry breaking, wherein local PAR-2-dependent weakening of the actin cortex, together with mutual inhibition of anterior and posterior PAR proteins, provides a mechanism for spontaneous symmetry breaking without centrosomes.
DOI: https://doi.org/10.7554/eLife.44552.001

*For correspondence:
pierre.gonczy@epfl.ch

**Competing interests:** The authors declare that no competing interests exist.

## Introduction

Symmetry breaking is a fundamental feature of living systems, operating at different scales and in different contexts. One particularly critical instance occurs at the onset of development, when symmetry must be broken in a tightly coordinated manner to ensure correct specification of body axes. The mechanisms through which this is achieved in an organized manner remain incompletely understood.

The zygote of the nematode *Caenorhabditis elegans* is a powerful model system for dissecting the mechanisms that govern symmetry breaking and anterior-posterior (A-P) axis specification (reviewed in ***Rose and Gonczy, 2014***). Shortly after fertilization, the zygote undergoes the two female meiotic divisions, leading to extrusion of the polar bodies, usually at the future anterior pole. The cell cortex is isotropic at this early stage, as evidenced by uniform acto-myosin contractions driven by the small GTPase RHO-1. Thereafter, the RHO-1 guanine-nucleotide-exchange factor (GEF) ECT-2 is cleared from the cortex in the vicinity of the sperm-contributed centrioles, which are usually located on the opposite side from the polar bodies, leading to local RHO-1 inactivation and

cortical weakening (*Motegi and Sugimoto, 2006*; *Zonies et al., 2010*). Concomitantly, the posterior polarity proteins PAR-2 and PAR-1 become enriched at that location, whereas flows of the acto-myosin cortex transport the anterior PAR complex (PAR-3/PAR-6/PKC-3) away from this region, resulting in polarization along the A-P embryonic axis. Theoretical analysis based on measured rate constants indicates that polarization arises from coupling advective cortical flows with a PAR reaction-diffusion system, whereby anterior PAR complex components antagonize plasma membrane binding of posterior PAR proteins (*Goehring et al., 2011*).

In the absence of cortical flows, a partially redundant pathway can polarize the embryo along the A-P axis (*Motegi et al., 2011*). The available evidence indicates that this pathway relies on PAR-2 itself, through an ability to bind phosphoinositides and microtubules nucleated by the two centrosomes located at the future embryo posterior. This serves to shield PAR-2 from the antagonizing effects of the anterior complex component PKC-3. This second pathway also entails a positive feedback loop of membrane-bound PAR proteins onto dissociation rates of the non-muscle myosin NMY-2, which can induce weak cortical flows without a posterior trigger such as that provided by the centrosomes (*Gross et al., 2019*). Furthermore, zygotes arrested for several hours in metaphase of the first meiotic division, owing to a mutation in the anaphase-promoting complex/cyclosome (APC/C) component MAT-1, recruit PAR-2 to the plasma membrane in the vicinity of the meiotic spindle, presumably due to the persistence of astral microtubules (*Wallenfang and Seydoux, 2000*). This indicates that at the least under some circumstances other areas of the cell cortex are competent to recruit PAR-2. In the wild type, both local RHO-1 inactivation and PAR-2-dependent pathways are thought to rely on centrosomes. This inference is based primarily on experiments in which centrosomes were ablated prior to polarity establishment using a laser microbeam, which resulted in the absence of PAR-2 from the entire cell cortex (*Cowan and Hyman, 2004*).

Centrosomes assemble in the *C. elegans* zygote around the pair of centrioles contributed by the sperm, in a manner that depends on the coiled-proteins SPD-2 and SPD-5 (*Hamill et al., 2002*; *Kemp et al., 2004*). Centrosome maturation occurs thereafter during the first cell cycle, and entails notably recruitment to the pericentriolar material (PCM) of the Aurora A kinase AIR-1 and then of γ-tubulin (TBG-1) (*Toya et al., 2011*). Both AIR-1 and TBG-1 are required for full microtubule organizing center (MTOC) activity of centrosomes, and thus for bipolar spindle assembly and chromosome segregation (*Hannak et al., 2001*; *Schumacher et al., 1998*; *Toya et al., 2011*). Despite the postulated essential role of centrosomes in A-P polarity and the knowledge about components critical for assembling them, the mechanisms through which centrosomes instruct symmetry breaking in *C. elegans* remain unclear. Moreover, whether distinct polarization mechanisms might exist in zygotes deprived of centrosomes from the very onset of development has not been investigated.

## Results

### AIR-1 ensures uniqueness of symmetry breaking in *C. elegans* zygotes

While examining the function of AIR-1 in spindle positioning (*Kotak et al., 2016*), as noted also previously (*Noatynska et al., 2010*; *Schumacher et al., 1998*), we observed that the majority of zygotes in which AIR-1 had been depleted by RNAi exhibited a remarkable polarity phenotype, with two distinct PAR-2 domains, one at each pole (see below). We set out to investigate this phenotype in depth and thus utilize it as a handle to decipher fundamental principles of symmetry breaking and A-P polarization in *C. elegans*.

We conducted time-lapse microscopy of embryos simultaneously expressing endogenously tagged fluorescent proteins marking NMY-2 to monitor the cortical acto-myosin network, SAS-7 to track centrosomes and PAR-2 to assay posterior polarization. As anticipated, by the time of pronuclear meeting, all control embryos established a single PAR-2 domain on the side of the centrosomes and the male pronucleus (*Figure 1A and C*, *Video 1*). In stark contrast,~76% *air-1*(*RNAi*) embryos formed two PAR-2 domains, one at each pole (*Figure 1B and C*, *Video 2*, referred to as 'bipolar'). Another ~19% *air-1*(*RNAi*) embryos established a single PAR-2 domain, usually on the side of the maternal pronucleus (*Figure 1C*, referred to as 'anterior') or else on that of the male pronucleus (*Figure 1C*, referred to as 'posterior'). Moreover,~5% *air-1*(*RNAi*) embryos harbored no plasma membrane PAR-2 (*Figure 1C*, referred to as 'none'). Analogous distributions were found for endogenous PAR-2 by immuno labeling wild type embryos subjected to RNAi-mediated depletion of AIR-1 (*Figure 1D*; *Figure 1—figure*

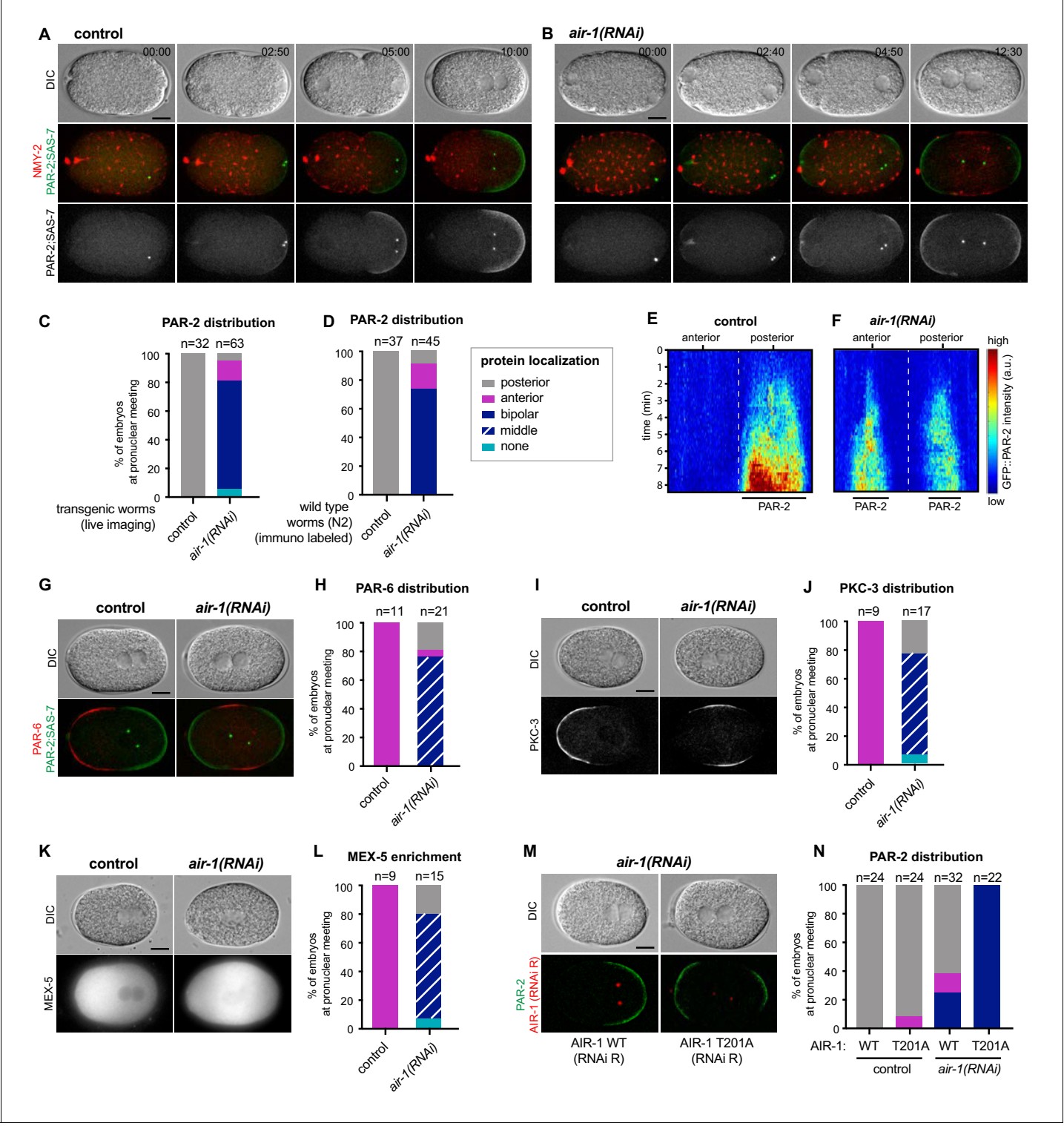

**Figure 1.** AIR-1 ensures uniqueness of symmetry breaking in *C.elegans* zygotes. (**A,B**) Time lapse microscopy of control (**A**) and *air-1(RNAi)* (**B**) embryos expressing RFP::NMY-2 (red), GFP::PAR-2 and GFP::SAS-7 (both green). Upper panels: DIC, middle panels: merge, lower panels: grey scale of GFP:: PAR-2 and GFP::SAS-7 signals. In this and all other panels, scale bar is 10 µM and time is shown in min:s. (**C**) Quantification of GFP::PAR-2 distributions at pronuclear meeting, which is the stage used in all subsequent panels. See ***Supplementary file 2*** for all statistical analyses (Fischer's exact test). (**D**) Quantification of endogenous PAR-2 distributions in wild type (N2) embryos. (**E,F**) Heat map of GFP::PAR-2 fluorescence over time along the circumference of control (F, n = 12) and *air-1(RNAi)* (G, n = 14; bipolar only) embryos. Tick marks above the panels indicate anterior-most and posterior-most positions, respectively. t = 0 corresponds to the onset of cortical flows (G) Control and *air-1(RNAi)* embryos expressing mCherry::PAR-2 (green),

*Figure 1 continued on next page*

Figure 1 continued

together with GFP::SAS-7 and GFP::PAR-6 (both red). Upper panels: DIC, lower panels: merge. (H) Quantification of GFP::PAR-2 distributions corresponding to G. (I,K) Control and *air-1(RNAi)* embryos expressing GFP::PKC-3 (I), mCherry::MEX-5 (K). Upper panels: DIC, lower panels: grey scale. (J,L) Quantification of GFP::PAR-2 distributions corresponding to I and K. (M) Images of embryos depleted of endogenous AIR-1 and expressing mCh::PAR-2 and RFP::SAS-7 (both red), together with GFP::AIR-1 WT(RNAi resistant, denoted RNAi R) (left) or GFP::AIR-1 T201A(RNAi R) (right) (both green). Upper panels: DIC, lower panels: merge. (N) Quantification of GFP::PAR-2 distributions in embryos expressing GFP::AIR-1 WT(RNAi R) or GFP::AIR-1 T201(RNAi R) without or with endogenous AIR-1 depletion, as indicated.

DOI: https://doi.org/10.7554/eLife.44552.002

The following figure supplements are available for figure 1:

**Figure supplement 1.** Aspects of polarity in AIR-1-depleted embryos.
DOI: https://doi.org/10.7554/eLife.44552.003
**Figure supplement 2.** PAR-2 domain distributions in different strains and conditions.
DOI: https://doi.org/10.7554/eLife.44552.004

*supplement 1A*). Note that we never detected an anterior PAR-2 domain either early or later in the cell cycle in the endogenously tagged GFP::PAR-2 strain used in this study, in contrast to the situation in strains overexpressing GFP::PAR-2 (*Figure 1—figure supplement 2*). We noted also that progressive dilutions of *air-1(RNAi)* resulted in weaker spindle assembly defects and shifted the polarity phenotype away from harboring two poles (*Figure 1—figure supplement 1B,C*). Therefore, bipolarity is the most severe AIR-1 depletion phenotype obtained using RNAi, likely corresponding to a strong if not complete loss of function in the embryo. Note that null alleles of *air-1* result in sterility (*Furuta et al., 2002*), precluding their analysis in the embryo. Overall, we conclude that AIR-1 ensures the establishment of a unique posterior PAR-2 region in the *C. elegans* zygote.

We set out to analyze further features of PAR-2 domain formation in *air-1(RNAi)* embryos. We found that the fluorescence intensity of each of the two PAR-2 domains was weaker than that of the single PAR-2 domain in control embryos (*Figure 1E,F*; *Figure 1—figure supplement 1F*). Likewise, each of the two PAR-2 domains was smaller (*Figure 1—figure supplement 1E*). However, the summed fluorescence intensities and sizes of both PAR-2 domains did not differ significantly from that of the single PAR-2 domain in control embryos (*Figure 1—figure supplement 1E,F*). To investigate whether *air-1(RNAi)* embryos underwent *bona fide* bipolarization, and not merely aberrant distribution of PAR-2, we examined the distribution of other PAR proteins. Both PAR-6 and PKC-3 localized to the anterior pole in all control embryos but were restricted to the middle region in ~80% *air-1(RNAi)* embryos, as anticipated from the bipolar distribution of PAR-2 (*Figure 1G–J*). Furthermore, we found that the cytoplasmic polarity mediator protein MEX-5, which was enriched in the anterior of control embryos in response to proper A-P polarity (*Schubert et al., 2000*), was present primarily in the middle region of ~80% *air-1(RNAi)* embryos (*Figure 1K,L*). These findings indicate that the machinery that ensures polarization operates normally in *air-1(RNAi)* embryos, but that its spatial regulation is impaired.

To determine whether AIR-1 kinase activity is required for correct polarization, we analyzed embryos expressing GFP fusion proteins to either a wild type or a kinase inactive version of RNAi-resistant AIR-1 (*Toya et al., 2011*). We found that whereas wild type GFP::AIR-1 rescued posterior PAR-2 polarity upon depletion of endogenous AIR-1 in ~63% of embryos, the kinase inactive version did not, with all embryos being bipolar (*Figure 1M,N*). The fact that all embryos were bipolar in this case, compared to ~76% in plain *air-1(RNAi)* embryos, likely reflects the partial functional impairment that is also observed in the wild type construct (see *Figure 1N*), which may result from the presence of mCherry or else from altered expression levels. Overall, we conclude that AIR-1 kinase activity is required to ensure a unique symmetry breaking event in *C. elegans* zygotes. Whereas the identification of relevant AIR-1 substrates will be of interest for future investigations, hereafter we utilize this

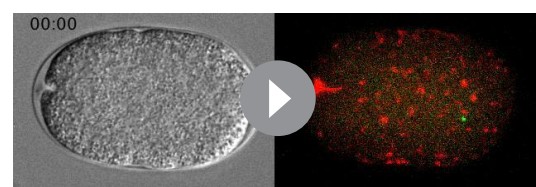

**Video 1.** Control embryo expressing RFP::NMY-2 (red); GFP::PAR-2 and GFP::SAS-7 (both green). Left: DIC, right: merge.
DOI: https://doi.org/10.7554/eLife.44552.005

**Video 2.** *air-1(RNAi)* embryo expressing RFP::NMY-2 (red); GFP::PAR-2 and GFP::SAS-7 (both green). Left: DIC, right: merge.
DOI: https://doi.org/10.7554/eLife.44552.006

remarkable bipolar phenotype to decipher fundamental principles underlying symmetry breaking at the onset of A-P axis specification in *C. elegans.*

## Bipolarity of embryos depleted of AIR-1 is established through the PAR-2 pathway

Since polarity establishment normally relies in a partially redundant manner on local RHO-1 inactivation mediated by cortical flows and on the PAR-2-dependent pathway (*Motegi et al., 2011*), we investigated these mechanisms in *air-1(RNAi)* embryos to assess whether their alteration could be responsible for imparting bipolarity. Examining cortical flows, we found that control embryos exhibited the expected anterior-directed movements of NMY-2 emanating from the centrosomes-bearing region, followed by the appearance of a posterior PAR-2 domain (*Figure 2A*). By contrast, embryos depleted of AIR-1 exhibited cortical flows stemming from both poles and directed toward the embryo center, followed by PAR-2 domain establishment at both poles (*Figure 2B*). Quantification of cortical flows using particle image velocimetry (PIV) of RFP:: NMY-2 revealed that the average peak velocities were ~6.8 μm/min in control embryos, but ~3.1 μm/ min for posterior- and ~2.9 μm/min for anterior-directed flows, respectively, in *air-1(RNAi)* embryos (*Figure 2B,D*). We conclude that AIR-1 is somehow required for robust cortical flows.

We set out to test whether the decrease in cortical flow velocity in *air-1(RNAi)* embryos contributes to the bipolar phenotype. We reasoned that if this were the case, then increasing flow velocities in *air-1(RNAi)* embryos should rescue the phenotype. To this end, we set out to co-deplete AIR-1 and the RHO-1 GTPase activating protein (GAP) RGA-3 (*Schonegg et al., 2007*). On its own, RGA-3 depletion led to stronger cortical flows and to an enlarged posterior PAR-2 domain (*Figure 2—figure supplement 1A,E*) (*Schonegg et al., 2007*). We found that the average peak velocities of the cortical flows in doubly depleted *air-1(RNAi) rga-3(RNAi)* embryos were close to those of control embryos (*Figure 2—figure supplement 1B*). Despite this, the fraction of *air-1(RNAi) rga-3(RNAi)* embryos exhibiting a bipolar phenotype was not altered (*Figure 2E,F*). Therefore, while cortical flows are diminished upon AIR-1 depletion, this alone does not explain the bipolar phenotype. To investigate whether cortical acto-myosin flows present in *air-1(RNAi)* embryos are required for bipolarity, we set out to utilize *nmy-2(ne3409)* embryos. This temperature-sensitive mutant allele of NMY-2 exhibits severely reduced cortical flows (*Liu et al., 2010*), to an extent similar to those observed on either side of *air-1(RNAi)* embryos (*Figure 2—figure supplement 1C*). We found that *nmy-2(ne3409)* embryos did not exhibit bipolarity and instead harbored a single posterior PAR-2 domain, which formed later and was smaller than in the control (*Figure 2G,H*; *Figure 2—figure supplement 1E*), in line with prior work (*Liu et al., 2010*). Depletion of AIR-1 in *nmy-2(ne3409)* embryos almost completely abolished cortical flows (*Figure 2—figure supplement 1C*), but small bipolar PAR-2 domains formed nevertheless in most embryos, although with a substantial delay (*Figure 2G, H*; *Figure 2—figure supplement 1F*). Together, these findings indicate that cortical acto-myosin flows present in *air-1(RNAi)* embryos are not essential for bipolarity, but contribute to the robustness and extent of the two PAR-2 domains.

The above observations prompted us to test whether the bipolar phenotype relies on the PAR-2-dependent pathway (*Motegi et al., 2011*). To this end, we used a strain that expresses an RNAi-resistant GFP-tagged version of PAR-2, either in the wild type (WT) form or in mutant forms deficient for PAR-2 binding to microtubules (PAR-2 R163A or PAR-2 R183-5A) and thus lacking the second polarity pathway. As reported previously (*Motegi et al., 2011*), whereas all three transgenes rescued depletion of endogenous PAR-2 to almost 100%, diminishing cortical flows by co-depletion of ECT-2 abolished loading of the two PAR-2 mutants onto the plasma membrane (*Figure 2I–K*; *Figure 2—figure supplement 1G–J*). Strikingly, we found the same to be true in these mutants when AIR-1 was co-depleted with endogenous PAR-2 (*Figure 2I–K*; *Figure 2—figure supplement 1G–J*). Moreover, whereas cortical flows were normal when PAR-2 alone was depleted (*Munro et al., 2004*) (*Figure 2—figure supplement 1K,M,N*), we found that they were absent when AIR-1 was depleted concomitantly (*Figure 2—figure supplement 1L,M,O*), reflecting the feedback loop of membrane-

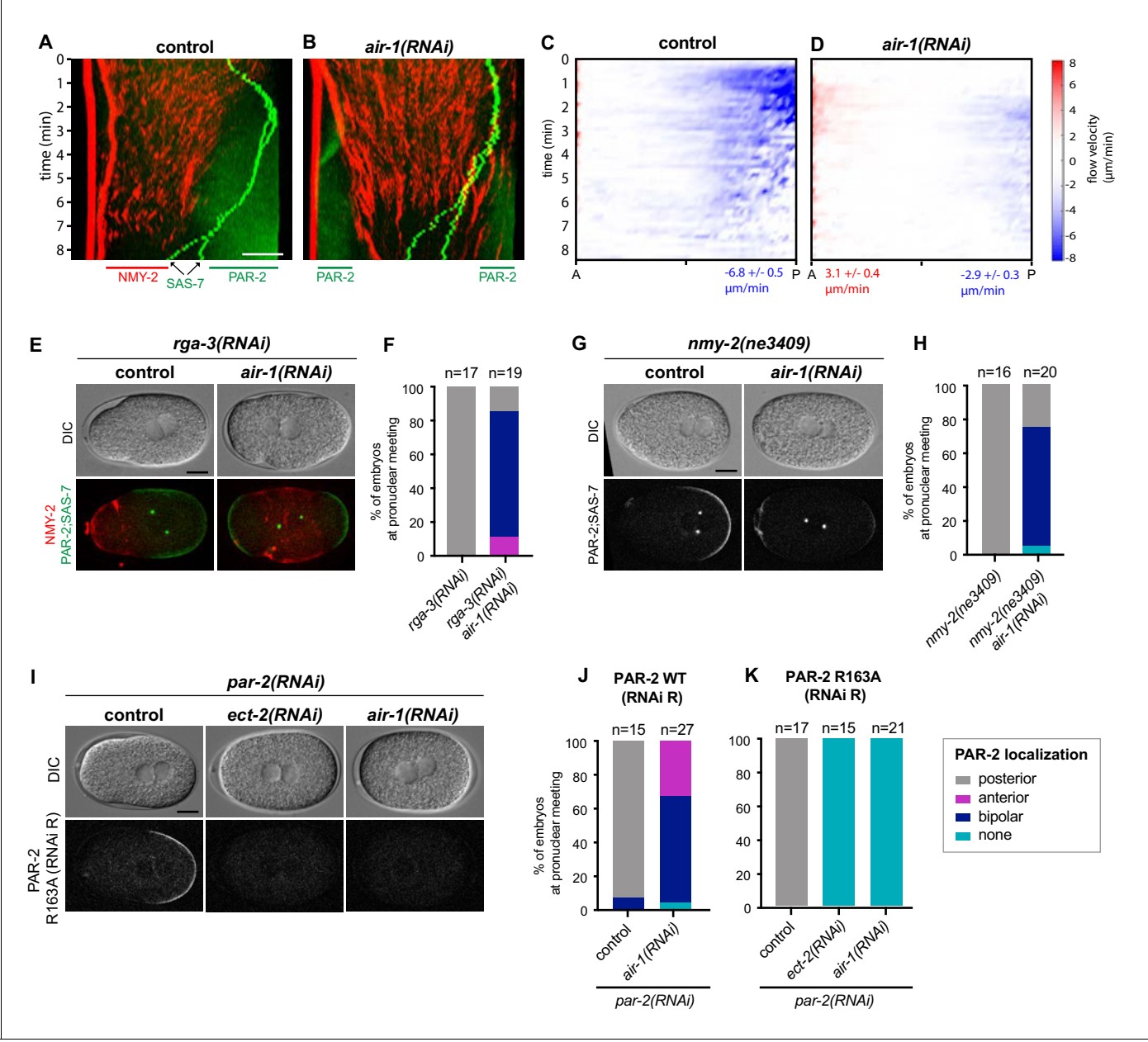

**Figure 2.** Bipolarity of embryos depleted of AIR-1 is established through PAR-2. (**A,B**) Kymographs of control (**A**) and *air-1(RNAi)* (**B**) embryos expressing RFP::NMY-2 (red), GFP::PAR-2 and GFP::SAS-7 (both green). Straight red vertical lines correspond to immobile polar bodies. (**C,D**) Kymograph of RFP::NMY-2 cortical flow velocities quantified by particle imaging velocimetry (PIV) in control (**C**) and *air-1(RNAi)* (**D**) embryos. Values indicate average peak velocities at the anterior (red) and posterior (blue). N = 10 for both conditions. (**E,G**) *rga-3(RNAi)* and *rga-3(RNAi) air-1(RNAi)* embryos (**E**) or *nmy-2(ne3409)* and *nmy-2(ne3409) air-1(RNAi)* (**G**) embryos expressing RFP::NMY-2 (red), GFP::PAR-2 and GFP::SAS-7 (both green). Upper panels: DIC, lower panel: merge. The black triangle in the lower left corner of the control embryo in (**G**) results from rotating the image. (**F,H**) Quantification of GFP::PAR-2 distributions corresponding to E and G. Note that GFP::PAR-2 domains were less robust in *nmy-2(ne3409) air-1(RNAi)* embryos than in the control condition (see G). (**I**) Embryos expressing GFP::PAR-2 R163A (RNAi-resistant, denoted RNAi R) depleted of endogenous PAR-2, together with *ect-2(RNAi)* or *air-1(RNAi)*, as indicated. Upper panels: DIC, lower panels: grey scale. (**J,K**) Quantification of GFP::PAR-2 distributions in embryos expressing RNAi-resistant GFP::PAR-2 WT or GFP::PAR-2 R163A and depleted of endogenous PAR-2 in combination with *ect-2(RNAi)* or *air-1(RNAi)*, as indicated.

DOI: https://doi.org/10.7554/eLife.44552.007

The following figure supplement is available for figure 2:

**Figure supplement 1.** Cortical flows in AIR-1 depleted embryos are PAR-2-dependent.

DOI: https://doi.org/10.7554/eLife.44552.008

bound PAR-2 onto cortical flows (*Gross et al., 2019*). Taken together, these results demonstrate that the bipolarity of *air-1(RNAi)* embryos relies on the PAR-2-dependent pathway.

## PAR-2 domains form preferentially in curved regions of the embryo

As PAR-2 domains developed invariably at the poles of *air-1(RNAi)* embryos, we reasoned that curvature might be a contributing cue for establishing PAR-2 domains. To test this hypothesis, we used PDMS to microfabricate equilateral triangular chambers ~ 40 μm in side length and ~25 μm in depth. We reasoned that PAR-2 domains should form preferentially in the corners of such triangular geometries upon AIR-1 depletion if curvature is indeed a contributing factor. In this set of experiments, embryos were squeezed into the triangular chambers prior to symmetry breaking and filming. Despite being deformed, control embryos established a single PAR-2 domain in the vicinity of centrioles, which in ~50% of cases were along one side of the triangle and in the remaining ~50% in one of the corners (*Figure 3A,B*, *Video 3*). Approximately 20% of *air-1(RNAi)* embryos formed two PAR-2 domains, with the remaining ~80% forming one domain (*Figure 3A,B*, *Videos 4* and *5*). Remarkably, with one exception, PAR-2 domains in *air-1(RNAi)* embryos formed within a corner of the triangle (*Figure 3C*), usually in the corner where the embryo exhibited the highest degree of curvature before symmetry breaking (*Figure 3D*, *Figure 3—figure supplement 1A*). Moreover, as shown in *Figure 3E*, PAR-2 domains in *air-1(RNAi)* embryos formed independently of the position of centrosomes or polar bodies, in contrast to control embryos, in which PAR-2 domain formation coincided simply with centrosome position. Together, these experiments indicate that embryos depleted of AIR-1 recruit PAR-2 preferentially to curved regions.

We investigated whether the preference of PAR-2 for curved regions stems from an accumulation of microtubules in these locations, which could conceivably enhance PAR-2 recruitment. However, immunofluorescence analysis did not reveal an accumulation of microtubules at the poles of *air-1* (*RNAi*) embryos (*Figure 3—figure supplement 1B,C*). Likewise, analysis of embryos expressing YFP::TBB-2 and placed vertically into cylindrical PDMS wells for end-on time lapse microscopy confirmed that there was no difference in microtubules distribution at poles when comparing control and *air-1(RNAi)* embryos (*Figure 3—figure supplement 1D,E*). Nevertheless, we tested whether microtubules might mediate PAR-2 recruitment in *air-1(RNAi)* embryos. We depleted microtubules shortly after fertilization using nocodazole, which resulted in a failure of female pronuclear migration (*Figure 3—figure supplement 1F*, arrow). As previously reported (*Hayashi et al., 2012*; *Sonneville and Gönczy, 2004*; *Tsai and Ahringer, 2007*), removing microtubules from control embryos did not prevent posterior PAR-2 domain establishment (*Figure 3—figure supplement 1F, G*). Importantly, nocodazole-treated *air-1(RNAi)* embryos recruited PAR-2 to both poles (*Figure 3—figure supplement 1F,G*). We found an analogous result in embryos lacking microtubules following RNAi-mediated depletion of the α-tubulin TBA-2 (*Figure 3—figure supplement 1H*). Overall, we conclude that the preference of PAR-2 for the polar regions of *air-1(RNAi)* embryos is not due to microtubule enrichment in those areas.

## Theoretical analysis of symmetry breaking in *C. elegans* zygotes

We developed a physical description of curvature-dependent symmetry breaking and PAR-dependent axis specification to better understand how the *C. elegans* zygote undergoes bipolarization when AIR-1 is lacking (*Figure 4A*, Appendix). Our 1D dynamical system is based on known interactions between anterior and posterior PAR proteins, which were also considered in a previous theoretical analysis (*Goehring et al., 2011*). In our system, the concentrations $A$ of membrane-bound anterior (PAR-6) and $P$ of posterior (PAR-2) proteins follow the dynamic equations

$$\partial_t A = D_A \partial_x^2 A - \partial_x(vA) + k_{on,A} A_{cyto} - k_{off,A} A - k_{AP} PA \tag{1}$$

$$\partial_t P = D_P \partial_x^2 P - \partial_x(vP) + k_{on,P} P_{cyto} - k_{off,P} P - k_{PA} A^2 P, \tag{2}$$

where $D_A$ and $D_P$ denote the respective diffusion coefficients of the two species, $v$ the velocity of the cortical flow, $k_{on}$ and $k_{off}$ the respective attachment and detachment rates, and $A_{cyto}$ and $P_{cyto}$ their cytoplasmic concentrations. The terms $k_{AP} PA$ and $k_{PA} A^2 P$ account for the mutual antagonism between anterior and posterior PAR proteins. In previous work, symmetry breaking was achieved by

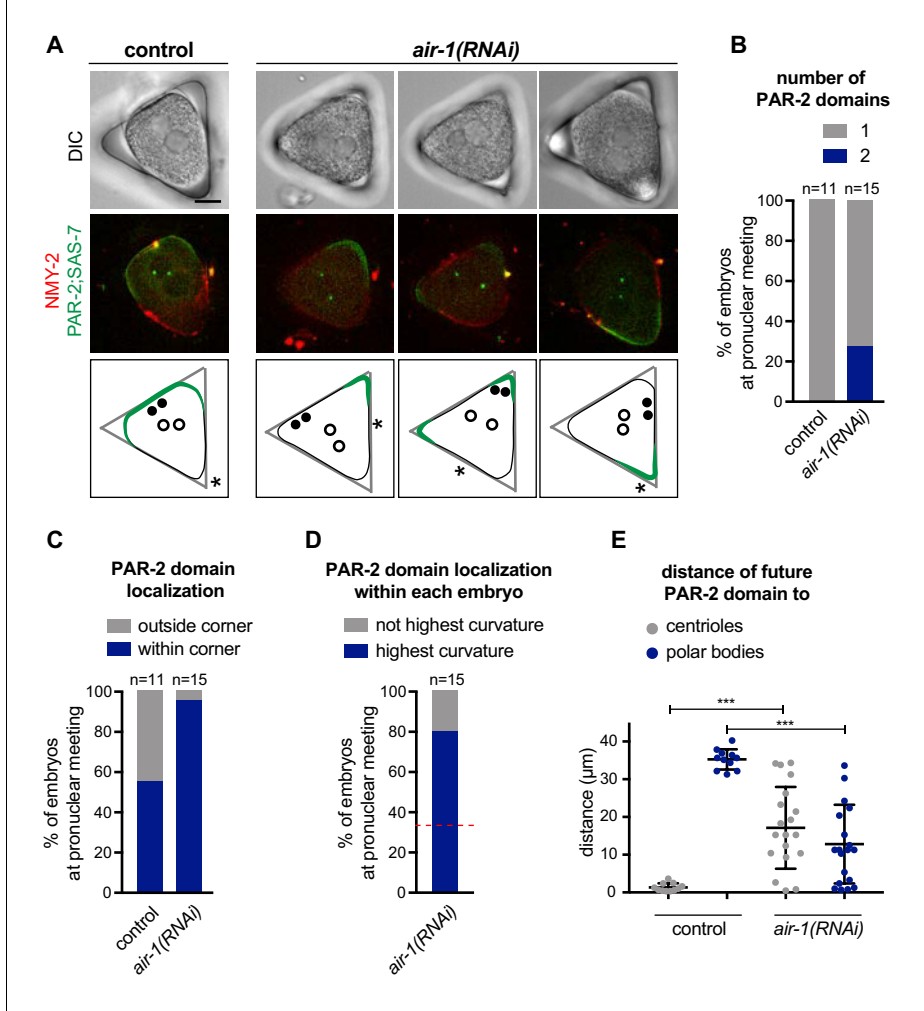

**Figure 3.** PAR-2 domains form in regions of high membrane curvature. (**A**) One control and three *air-1(RNAi)* embryos expressing RFP::NMY-2 (red), GFP::PAR-2 and GFP::SAS-7 (both green) placed in triangular PDMS chambers ~40 μM in side length. Lower panels illustrate the localization of PAR-2 domains at pronuclear meeting (green), of centrioles (filled discs: before symmetry breaking; empty circles: at pronuclear meeting) and of polar bodies (asterisks, before symmetry breaking). (**B–E**) GFP::PAR-2 distribution in control and *air-1(RNAi)* embryos in triangular chambers. (**B**): number of domains; (**C**): localization outside (no GFP::PAR-2 within any corner) or within a corner; (**D**): localization within the corner with the highest curvature or not; red dashed line indicates chance occurrence. The extent of curvature of the embryo *per se* was scored before symmetry breaking; (**E**): localization with respect to the position of centrioles and polar bodies before symmetry breaking.

DOI: https://doi.org/10.7554/eLife.44552.009

The following figure supplement is available for figure 3:

**Figure supplement 1.** PAR-2 domains form independently of microtubules.

DOI: https://doi.org/10.7554/eLife.44552.010

imposing a cortical flow from the posterior to the anterior side, which was assumed to be induced by centrosomes via an unspecified mechanism (*Goehring et al., 2011*). Here, we use instead active gel theory to include a physical mechanism that causes cortical flows (*Kruse et al., 2004*). In this approach, cortical flows are caused by gradients in the mechanical stress that is generated in the actin cortex notably through non-muscle myosin activity (*Callan-Jones et al., 2016*; *Callan-Jones and Voituriez, 2013*; *Gross et al., 2019*; *Hannezo et al., 2015*), following the force-balance relation

$$\eta \partial_x^2 v = \partial_x \left( \Pi_{active} + \Pi_{passive} \right) + \gamma v \tag{3}$$

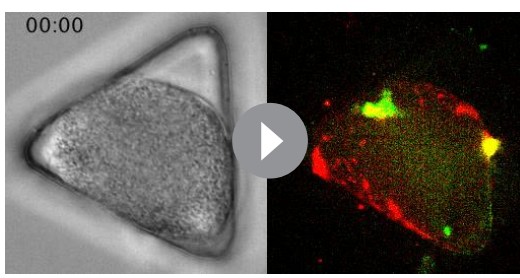

**Video 3.** Control embryo placed in a triangular chamber expressing RFP::NMY-2 (red); GFP::PAR-2 and GFP::SAS-7 (both green). Left: DIC, right: merge of fluorescent signals.
DOI: https://doi.org/10.7554/eLife.44552.011

In this equation, $\eta$ denotes the viscosity of the cortex and $\gamma$ its friction against the membrane. $\Pi_{active}$ accounts for the active stress, leading to contraction using molecular motors, and $\Pi_{passive}$ for the passive stress originating from osmotic pressure. It has been shown that dissociation rates of cortical NMY-2 are regulated by PAR proteins, with lower dissociation rates occurring in the PAR-6-containing domain than in the PAR-2-containing domain (*Gross et al., 2019*). To account for this feedback loop, we assume that the active stress $\Pi_{active}$ effectively decreases with the density of membrane-bound PAR-2 $P$ as $\beta P$ via antagonism with PAR-6. Furthermore, we account for the preferential binding of PAR-2 to curved regions without assigning a specific molecular mechanism. This feature is incorporated by increasing the rate $k_{on,P}$ of PAR-2 binding at the poles by an amount $\alpha$ multiplied by the excess curvature compared to the central parts of the cortex (see Appendix for a detailed description of the physical model).

We first analyzed our dynamical system in the wild type situation. In this case, we capture the impact of centrosomes through an initial condition with reduced cortical active stress in their vicinity. The corresponding phase diagram exhibits a large zone of a single PAR-2 domain (*Figure 4B*, yellow area). Using moderate values of $\alpha$ and $\beta$ in this parameter space (*Figure 4B*, red arrows), the numerical simulation yielded a single PAR-2 domain (*Figure 4C*, *Figure 4—figure supplement 1A*, *Video 6*). Below a threshold of $\alpha$, no polar PAR-2 domain was formed (*Figure 4B*, brown area), whereas for a very large curvature dependence of PAR-2 binding $\alpha$ and a very strong reduction of the cortical active stress $\beta$, a bipolar phenotype was obtained (*Figure 4B*, pale blue area). These two minor regions of the phase diagram seem outside of the parameter space in the wild type, since such embryos never exhibit bipolarity or lack polarity. Interestingly, this dynamical system also predicted that the PAR-2 domain relaxes to the nearby pole even if the initial active stress reduction occurred laterally (*Figure 4—figure supplement 1B*). This has been observed experimentally following lateral sperm entry in control embryos (*Figure 4—figure supplement 1C, D*) (*Goldstein and Hird, 1996*) or upon displacement of the PAR-2 domain from the pole using light-induced cytoplasmic streaming (*Mittasch et al., 2018*). This relaxation of the PAR-2 domain was also observed in *air-1(RNAi)* embryos, although in a less robust manner than in the wild type, likely due to impaired cortical flows (*Figure 4—figure supplement 1D-F*). Such congruence with experimental observations that were not included when developing our description provides further confidence that this dynamical framework provides a reliable representation of symmetry breaking and A-P polarization in *C. elegans*.

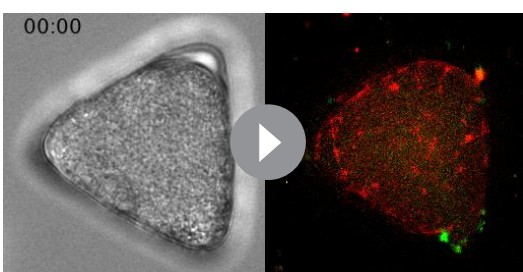

**Video 4.** *air-1(RNAi)* embryo placed in a triangular chamber expressing RFP::NMY-2 (red); GFP::PAR-2 and GFP::SAS-7 (both green). Left: DIC, right: merge of fluorescent signals.
DOI: https://doi.org/10.7554/eLife.44552.012

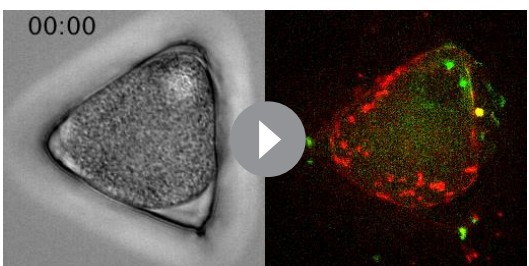

**Video 5.** Another example of an *air-1(RNAi)* embryo placed in a triangular chamber expressing RFP::NMY-2 (red); GFP::PAR-2 and GFP::SAS-7 (both green). Left: DIC, right: merge of fluorescent signals.
DOI: https://doi.org/10.7554/eLife.44552.013

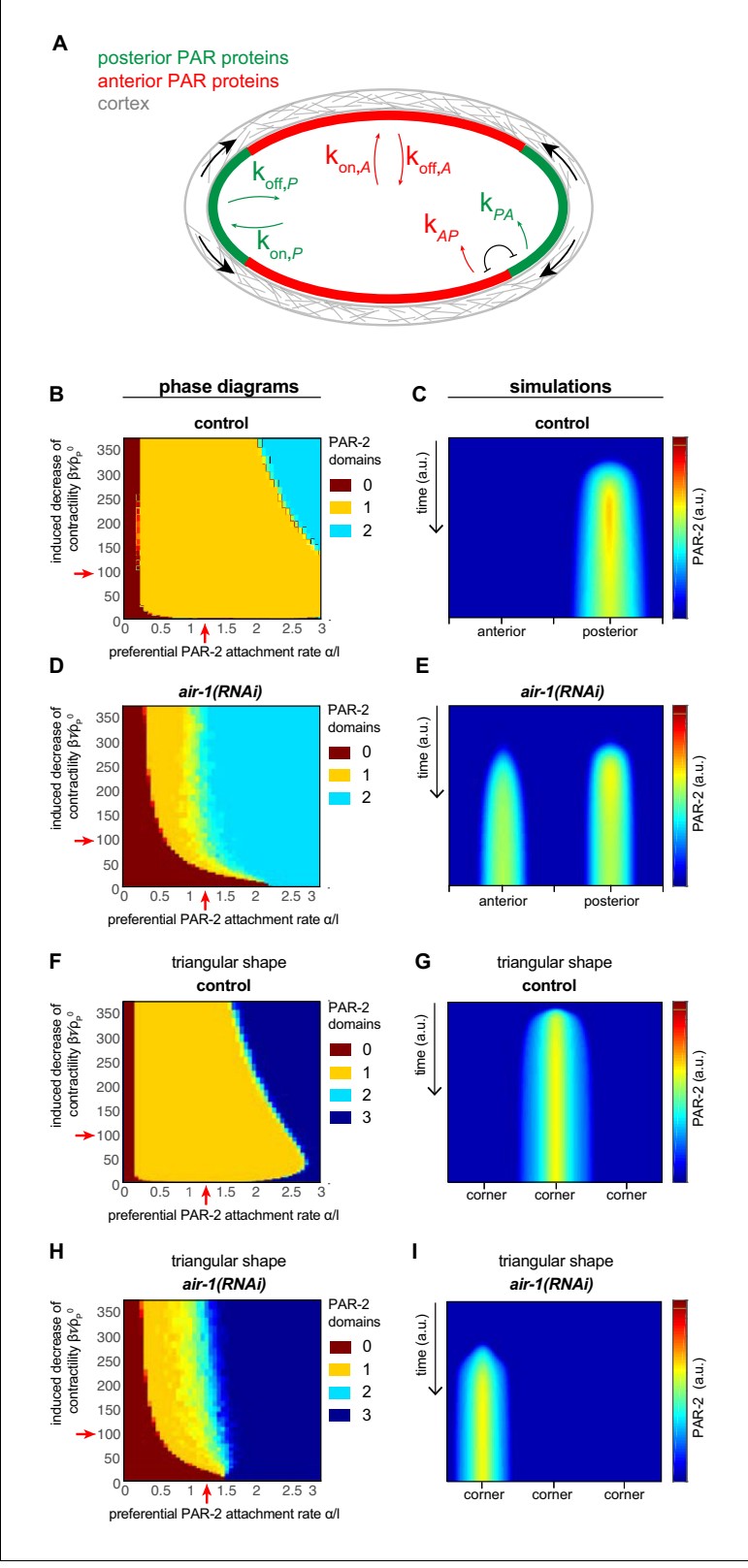

**Figure 4.** Physical description of spontaneous symmetry breaking. (**A**) Principal tenets of physical description of spontaneous symmetry breaking mechanism; see Appendix for details. (**B, D, F, H**) Phase diagrams showing the development of 0, 1 or 2 PAR-2 domains, as indicated, as a function of the preferential PAR-2 attachment rate $\alpha/l$ and the induced decrease of cortical contractility $\beta\tau/\rho_P^0$ in control (**B**) or *air-1(RNAi)* (**D**) embryos, as well as in

*Figure 4 continued on next page*

*Figure 4 continued*

control or *air-1(RNAi)* embryos in triangular chambers (F and H, respectively). (C, E, G, I) Simulation kymographs of PAR-2 distribution corresponding to the juxtaposed phase diagrams. Moderate values for $\alpha$ and $\beta$, away from the extremes, were chosen for the numerical simulation: $\alpha/l$=1.2 $\beta\tau/\rho_P^0$ 100.

DOI: https://doi.org/10.7554/eLife.44552.014

The following figure supplement is available for figure 4:

**Figure supplement 1.** Simulations of symmetry breaking and relaxation to the poles.

DOI: https://doi.org/10.7554/eLife.44552.015

Next, we analyzed this dynamical system in a situation where cortical active stress is not reduced in a given location by a localized centrosome-derived cue, as may be the case in *air-1(RNAi)* embryos. Strikingly, the corresponding phase diagram exhibited a massive increase in the region of bipolar PAR-2 domains and a significant region of coexisting polar and bipolar states (*Figure 4D*, pale blue and green areas). Using the same values for $\alpha$ and $\beta$ as in the wild type, the numerical simulation yielded two PAR-2 domains (*Figure 4E*; *Figure 4—figure supplement 1G*, *Video 7*). Furthermore, we solved the equations in a situation corresponding to wild type embryos in a triangular geometry, which yielded a phase diagram exhibiting a single PAR-2 domain, except for extreme values of $\alpha$ and $\beta$ (*Figure 4F*, yellow area). A corresponding numerical simulation is shown in *Figure 4G*. By contrast, the phase diagram reflecting the situation in *air-1(RNAi)* embryos placed in a triangular geometry exhibited a region of a single PAR-2 domain that was larger than that of regular *air-1(RNAi)* embryos (*Figure 4H*, yellow area, compare to *Figure 4D*), in line with the increased proportion of *air-1(RNAi)* embryos with just one such domain observed experimentally, and an abrupt transition to regions with two and then three PAR-2 domains (*Figure 4H*, pale and dark blue areas). The corresponding numerical solution with the same values for $\alpha$ and $\beta$ as above yielded a single PAR-2 domain (*Figure 4I*). Overall, we conclude that the physical description of symmetry breaking and PAR-dependent axis specification developed here can explain bipolarity of embryos in which cortical active stress is not reduced by a localized centrosome-derived cue.

## AIR-1 prevents symmetry breaking early in the cell cycle

We aimed to understand when and where in the embryo AIR-1 acts to ensure proper symmetry breaking and polarization. To this end, we first analyzed the distribution of endogenously GFP-tagged AIR-1 (*Sallee et al., 2018*). Intriguingly, we found that, before symmetry breaking, GFP::AIR-1 localized in the vicinity of the cell cortex (hereafter referred to as 'cortical'), as well as in the cytoplasm, and was barely detectable at centrosomes (*Figures 5A* and 00:00). Thereafter, the cortical and cytoplasmic pools decreased, which was paralleled by an increase of GFP::AIR-1 at maturing centrosomes (*Figures 5A* and 08:00-15:00). We found the same spatio-temporal distribution in wild-type embryos stained for endogenous AIR-1 (*Figure 5B*). Since the distribution of cortical AIR-1 is reminiscent of that of microtubules (*O'Rourke et al., 2010*), we assessed whether GFP::AIR-1 co-localized with mCherry::TBA-2, and found this to be the case indeed (*Figure 5C*). Likewise, endogenous AIR-1 co-localized with cortical microtubules before symmetry breaking in wild-type embryos (*Figure 5H*; *Figure 5—figure supplement 1A*). Importantly, we found that cortical AIR-1 was lost upon nocodazole-mediated depletion of microtubules (*Figure 5E,I*), whereas the centrosomal pool of AIR-1 was unaffected (*Figure 5—figure supplement 1B*). In contrast, depletion of AIR-1 did not alter the localization of cortical microtubules, but led to a slight increase in the corresponding signal intensity (*Figure 5D*, *Figure 5—figure supplement 1A*; *Figure 3—figure supplement 1A,B*). Overall, given that the absence of microtubules does not prevent A-P polarity (*Cowan and Hyman, 2004*; *Sonneville and Gönczy, 2004*; *Tsai and Ahringer, 2007*), these experiments lead us to propose that cortical AIR-1 does not modulate polarity establishment and that AIR-1 functions instead elsewhere, presumably in the cytoplasm, to prevent aberrant symmetry breaking early in the cell cycle.

Interestingly, we found that SPD-2 or SPD-5, which are both required to recruit AIR-1 to centrosomes (*Hamill et al., 2002*; *Kemp et al., 2004*), are also needed for the presence of AIR-1 at the cell cortex, as well as for limiting the intensity of the cortical microtubule network signal (*Figure 5F, G*; *Figure 5—figure supplement 1A*). To test whether AIR-1 kinase activity is needed for cortical

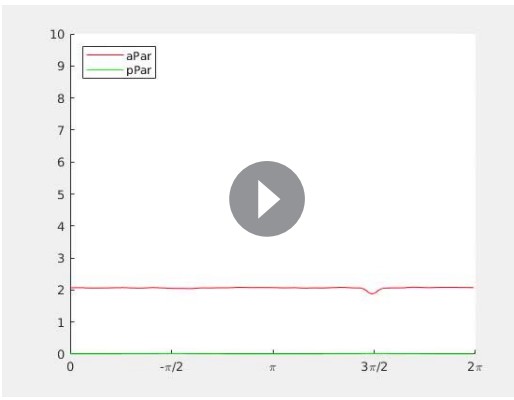

**Video 6.** Density of anterior and posterior PAR proteins (aPAR, pPAR) over time from solving the dynamical **Equations (1-3)** in a control embryo.
DOI: https://doi.org/10.7554/eLife.44552.016

localization early in the cell cycle, we again analyzed transgenic embryos expressing GFP fusion proteins to either a wild type or to a kinase inactive version of RNAi-resistant AIR-1 (**Toya et al., 2011**), depleting endogenous AIR-1 using RNAi. Although wild-type GFP::AIR-1 was both cortical and centrosomal, as anticipated, the kinase inactive version did not localize to the cortex and barely to the centrosomes (**Figure 5J,K**). Overall, we conclude that microtubules, SPD-2, SPD-5 and AIR-1 kinase activity are needed for the presence of AIR-1 at the cell cortex early in the cell cycle.

To explore whether the cortical pool of AIR-1 might be sufficient to modulate A-P polarity, we utilized a strain expressing the GFP binding protein (GBP) fused to mCherry and to RNAi-resistant AIR-1 as a mean to localize AIR-1 at will to GFP-positive cellular compartments (**Rothbauer et al., 2008**). The GBP::mCherry:: AIR-1 fusion protein rescued depletion of endogenous AIR-1 in ~67% of embryos, demonstrating that it possesses substantial activity (**Figure 6B,C**). As anticipated, co-depleting endogenous AIR-1 and SPD-2 in the presence of GBP::mCherry::AIR-1 resulted primarily in bipolar embryos as scored by mCherry::PAR-2 (**Figure 6B,C**). To target AIR-1 to the vicinity of the cell cortex, we crossed worms expressing GBP::mCherry::AIR-1 with animals expressing GFP::PH, which binds to phosphatidylinositol 4,5-bisphosphate (PI(4,5)P$_2$) at the plasma membrane (**Figure 6A**) (**Audhya et al., 2005**). This yielded a 2.5 fold increase of AIR-1 in the vicinity of the plasma membrane compared to cortical GBP::mCherry::AIR-1 alone (**Figure 6—figure supplement 1A,B**). Expression of mCherry::PAR-2 served as a readout for polarization in these experiments, whereas simultaneous depletion of SPD-2 ensured that no AIR-1 was targeted to centrosomes (**Figure 6—figure supplement 1A**). Importantly, we found that recruiting GBP::mCherry::AIR-1 to the vicinity of the plasma membrane prevented PAR-2 domain formation in ~72% of embryos, demonstrating that an excess of cortical AIR-1 can prevent symmetry breaking (**Figure 6B,C**).

To test whether AIR-1 can also exhibit inhibitory function on symmetry breaking at endogenous protein levels, we made use of the temperature-sensitive mutant allele *mat-1(ax161),* which results in embryos being arrested in metaphase of meiosis I (**Shakes et al., 2003**). Such arrested embryos developed a small anterior PAR-2 domain when shifted to 24°C (**Figure 6D,E**; **Figure 6—figure supplement 1C**) (**Wallenfang and Seydoux, 2000**). Since AIR-1 distribution is cortical and cytoplasmic, but not centrosomal, during meiosis I (see **Figures 5A**), we assessed polarity in *mat-1 (ax161) air-1(RNAi)* embryos to further assess the role of non-centrosomal AIR-1. When shifted to 24°C for 2 hr,~19% of *mat-1(ax161)* embryos developed a PAR-2 domain on the anterior side (**Figure 6D,E**; **Figure 6—figure supplement 1C**), a percentage that climbed to ~67% after 12 hr at 24°C (6D,E; **Figure 6—figure supplement 1C**). Strikingly, we found that ~39% *mat-1(ax161) air-1(RNAi)* embryos developed an anterior PAR-2 domain after 2 hr, and ~79% after 12 hr (**Figure 6D,E**; **Figure 6—figure supplement 1C**). Moreover, PAR-2 domains were more robust in *mat-1(ax161) air-1(RNAi)* embryos compared to the *mat-1(ax161)* mutant alone

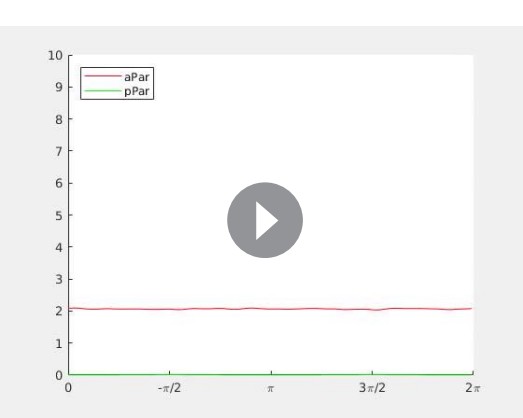

**Video 7.** Density of anterior and posterior PAR proteins (aPAR, pPAR) over time from solving the dynamical **Equations (1-3)** in a *air-1(RNAi)* embryo.
DOI: https://doi.org/10.7554/eLife.44552.017

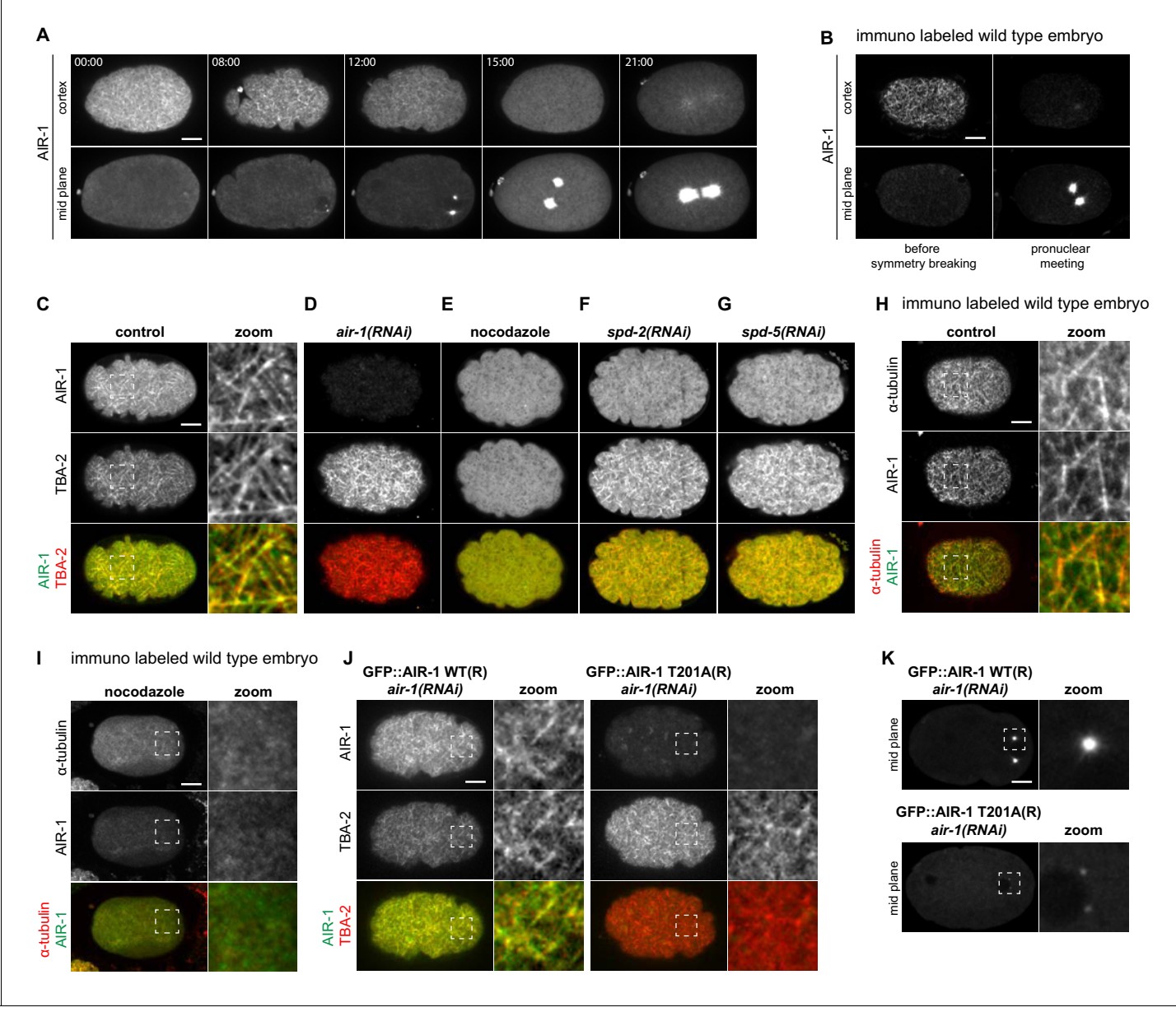

**Figure 5.** AIR-1 localizes to cortical microtubules. (**A**) Time lapse microscopy of an embryo expressing endogenously tagged GFP::AIR-1. Upper panels: maximum z-projection of 1.5 μm of the cortical planes. Lower panels: maximum z-projection of 7.5 μm of the middle planes of the same embryo. A representative embryo out of 9 that were imaged is shown. (**B**) Wild-type embryos fixed and stained for endogenous AIR-1 before symmetry breaking (left panels) and at pronuclear meeting (right panels). Upper panels: cortical plane; lower panels: middle plane of the same embryo. (**C–F**) Cortical images of embryos expressing endogenously tagged GFP::AIR-1 (green) and mCh::TBA-2 (red) prior to symmetry breaking; embryos were depleted by RNAi of AIR-1 (**D**), SPD-2 (**F**) or SPD-5 (**G**), or else treated with nocodazole to depolymarize microtubules (**E**). Depletion of AIR-1 (**D**), SPD-2 (**F**) or SPD-5 (**G**) by RNAi, and depletion of microtubules by nocodazole (**E**). The zoom highlights co-localization of GFP::AIR-1 with mCh::TBA-2 in the control condition. (**H,I**) Cortical images of wild-type (**H**) or nocodazole-treated (**I**) embryo fixed and stained for α-tubulin and AIR-1. The zoomed regions highlight co-localization of endogenous AIR-1 with microtubules in the control condition. (**J**) Embryos prior to symmetry breaking expressing GFP::AIR-1 WT or GFP::AIR-1 T201A (RNAi-resistant, denoted R) (green) and mCh::TBA-2 (red) and depleted of endogenous AIR-1 using RNAi. The zoomed regions highlight co-localization of α-tubulin and AIR-1 solely in the case of the wild type transgenic construct. Only the cortical plane is shown. (**K**) Cortical images of embryos expressing GFP::AIR-1 WT or GFP::AIR-1 T201A (RNAi resistant, denoted R) (green) and mCh::TBA-2 (red) at the onset of pronuclear migration. *air-1(RNAi)* is directed against endogenous AIR-1. (**C–K**) Representative images are shown out of >10 embryos analyzed for each condition.

DOI: https://doi.org/10.7554/eLife.44552.018

The following figure supplement is available for figure 5:

**Figure supplement 1.** Cortical AIR-1 co-localizes with cortical microtubules.

*Figure 5 continued on next page*

*Figure 5 continued*

DOI: https://doi.org/10.7554/eLife.44552.019

(*Figure 6—figure supplement 1C*). Next, we tested whether these findings could be captured by the dynamical system developed above. To this end, we translated the observed meiotic defects leading to anterior PAR-2 recruitment by imposing a weak anterior trigger (see Appendix). The resulting phase diagram representing *mat-1(ax161) air-1*(RNAi) embryos captured the experimentally observed shift from bipolarity towards a single PAR-2 domain (*Figure 6F*, yellow area, compare to *Figure 4D*). Taken together, these findings establish that endogenous levels of cortical or cytoplasmic AIR-1 can also negatively regulate polarity establishment upon compromised meiotic progression.

## The *C. elegans* zygote can undergo spontaneous polarization in a centrosome-independent manner

As a step towards investigating whether the centrosomal pool of AIR-1 contributes to proper symmetry breaking, we tested whether depleting other centrosomal components also gives rise to bipolar embryos. Importantly, we observed the same bipolar phenotype in mutant alleles of *spd-2* and *spd-5* (*Figure 7A,B*), as indicated previously (*Hamill et al., 2002*; *O'Connell et al., 2000*). By contrast, we found that depleting TBG-1, which is recruited to centrosomes in an AIR-1-dependent manner and is required for MTOC activity, does not impact polarity (*Figure 7A,B*) (*Hannak et al., 2001*).

Centrosomes are thought to be essential for symmetry breaking and PAR-2 recruitment based on the outcome of laser ablation experiments (*Cowan and Hyman, 2004*). Why, then, do *air-1*(RNAi) embryos establish two PAR-2 domains instead of none, as would have been expected in the absence of functional centrosomes? We noticed that, in contrast to control embryos, the position of centrioles in *air-1*(RNAi) embryos does not always match the location or PAR-2 domain initiation (*Figure 7A-Figure 7—figure supplement 1*). This mirrors the findings in the triangular chambers and suggests that bipolarization occurs in a centrosome-independent manner. To thoroughly address this possibility, we first assayed embryos derived from *such-1(t1668)* mutant sperm, a subset of which do not harbor DNA or centrioles owing to meiotic defects during spermatogenesis (*Figure 7C*) (*Bezler and Gönczy, 2010*). Embryos fertilized by such sperm are recognizable by the absence of a male pronucleus, of microtubule asters and of GFP::SAS-7 signal (*Figure 7D,E*). To further test whether these embryos lack centrioles and not merely GFP::SAS-7, we labeled *such-1 (t1668)* mutant embryos expressing GFP::PAR-2 and the centriolar marker GFP::SAS-4 with antibodies against GFP, as well as against AIR-1 as a proxy for centrosomes. As shown in *Figure 7—figure supplement 1B*, this experiment confirmed the absence of centrioles and centrosomes in embryos derived from *such-1(t1668)* mutant sperm. Strikingly, we found that such embryos usually establish a bipolar or an anterior PAR-2 domain (*Figure 7D,E*), indicating that centrosomes are dispensable for PAR-2 plasma membrane loading.

In an attempt to resolve the apparent discrepancy with the previous study (*Cowan and Hyman, 2004*), we conducted the same type of laser ablation experiment, but now using GFP::SAS-7 to monitor centrioles instead of GFP::SPD-2 as in the previous work. The bright GFP::SAS-7 signal enabled us to ablate centrosomes already during meiosis II. Importantly, we found that six out of nine embryos established bipolar GFP::PAR-2 domains upon centrosome ablation, with the remaining three developing an anterior PAR-2 domain (*Figure 7F–H*, *Video 8*). In contrast, embryos that underwent incomplete centrosome ablation and served as controls established a regular posterior PAR-2 domain (*Figure 7—figure supplement 1C*, *Video 9*). Although the above experiments further establish that centrosomes are dispensable for symmetry breaking, we considered whether sperm entry might impart a signal in the presumptive embryo posterior. To address this possibility, we took advantage of the fact that sperm occasionally enters the oocyte next to its nucleus (*Goldstein and Hird, 1996*). Importantly, we found that most *air-1*(RNAi) embryos were bipolar even in those cases where both male and female pronuclei were on the same side of the embryo (*Figure 7I,J*). Together, these experiments lead to the unequivocal conclusion that *C. elegans* embryos undergo spontaneous symmetry breaking in an unregulated manner in the absence of centrioles and centrosomes.

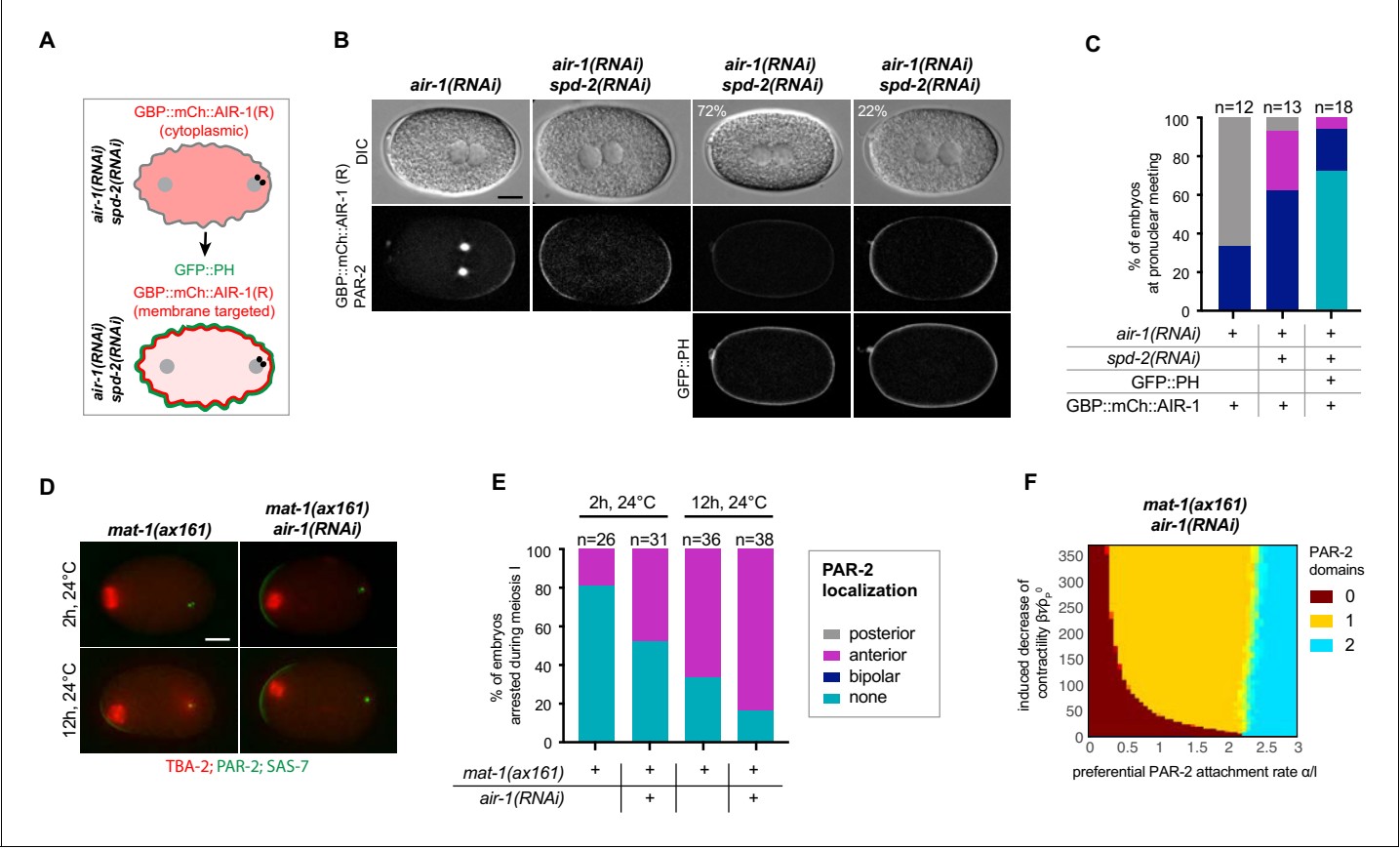

**Figure 6.** Cortical AIR-1 can act as a negative regulator of symmetry breaking. (**A**) Schematic illustrating the experimental set up to re-localize GBP::mCh::AIR-1 to the plasma membrane through binding to GFP::PH. Note that GBP::mCh::AIR-1(R) is not cortical in embryos depleted of SPD-2. (**B**) Embryos expressing GBP::mCh::AIR-1(RNAi resistant, denoted R), mCh::PAR-2 and GFP::PH depleted of endogenous AIR-1 and SPD-2. Note that the fluorescent intensity of mCh::PAR-2 is much stronger than that of membrane-targeted GBP::mCh::AIR-1(R), allowing us to unambiguously score polarity using the mCh::PAR-2 signal. (**C**) Quantification of mCh::PAR-2 distributions corresponding to (**B**). (**D**) *mat-1(ax161)* control and *mat-1(ax161) air-1(RNAi)* embryos expressing GFP::TBA-2 (red), mCh::PAR-2 and RFP::SAS-7 (both green) shifted to the restrictive temperature (24°C) for 2 hr or 12 hr to block them in metaphase of meiosis I. (**E**) Quantification of mCh::PAR-2 distributions corresponding to (**D**). (**F**) Phase diagram stemming from the physical description and showing the development of 0, 1 or 2 PAR-2 domains, as indicated, as a function of the preferential PAR-2 attachment rate $\alpha/l$ and the induced decrease of cortical contractility $\beta\tau/\rho_P^0$ in *mat-1(ax161) air-1(RNAi)* embryos.

DOI: https://doi.org/10.7554/eLife.44552.020

The following figure supplement is available for figure 6:

**Figure supplement 1.** Cortical AIR-1 can negatively regulate symmetry breaking.

DOI: https://doi.org/10.7554/eLife.44552.021

## Centrosomal AIR-1 is sufficient to induce symmetry breaking

We wanted to test whether AIR-1 is not only necessary but also sufficient to ensure a single symmetry breaking event. To address this question, we again made use of the strain expressing the relocalizable GBP::mCherry::AIR-1. Strikingly, we found that localizing GBP::mCherry::AIR-1 to centrioles by crossing GFP::SAS-7 in this strain rescued posterior polarity of mCherry::PAR-2 in the majority of embryos (*Figure 7K,L*), although the PAR-2 domain was smaller compared to the control condition (*Figure 7D*). Importantly, centrioles in this experiment do not possess a PCM since SPD-2 was depleted as well; as a result, such embryos do not form a spindle (*Figure 7—figure supplement 1E*). We conclude that AIR-1 localized on centrioles is sufficient to impart substantial posterior polarity.

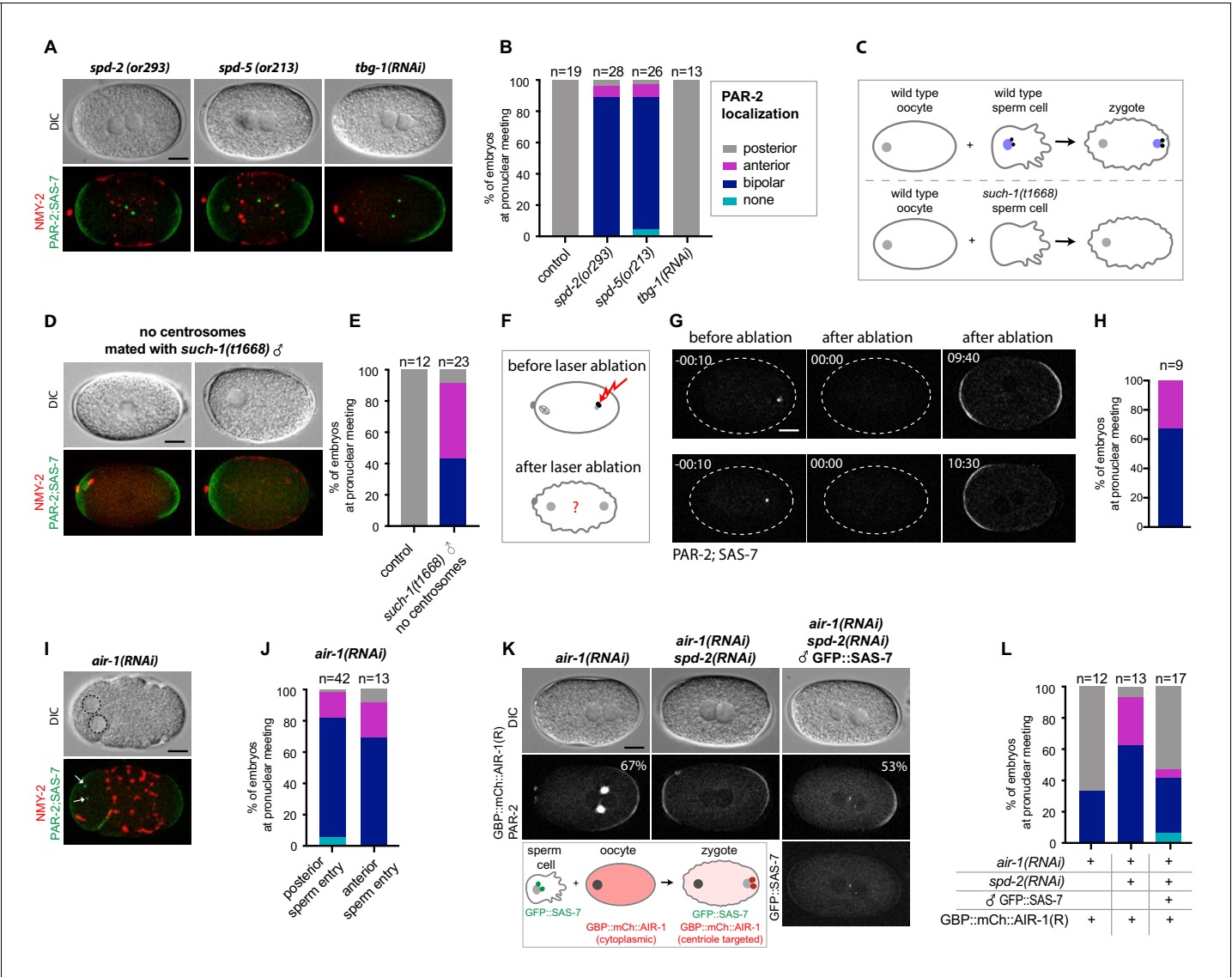

**Figure 7.** Centrosomes ensure uniqueness of symmetry breaking in *C.elegans* zygotes. (A) *spd-2(or293)*, *spd-5(or213)* or *tbg-1(RNAi)* embryos, as indicated, expressing RFP::NMY-2 (red), GFP::PAR-2 and GFP::SAS-7 (both green). Upper panels: DIC, lower panels: grey scale. (B) Quantification of GFP::PAR-2 distributions corresponding to (A). (C) Schematic showing experimental set up for (D) and (E). (D) *fem-1(hc17)* hermaphrodites expressing RFP::NMY-2 (red), GFP::PAR-2 and GFP::SAS-7 (both green) mated with *such-1(t1668)* males. Note lack of male pronucleus and of centriolar GFP::SAS-7. (E) Quantification of GFP::PAR-2 distributions corresponding to (D). (F) Schematic illustrating the experimental set-up of the centrosome laser ablation experiments. (G) Two embryos expressing GFP::PAR-2 and GFP::SAS-7 prior to laser ablation (t=-00:10), immediately after laser ablation of the centrosome (t = 00:00) and at pronuclear meeting (t = 09:40 and t = 10:30). (H) Quantification of GFP::PAR-2 distributions corresponding to (G). (I) *air-1 (RNAi)* embryo expressing RFP::NMY-2 (red), GFP::PAR-2 and GFP::SAS-7 (both green) with sperm entry next to the maternal pronucleus. Dotted lines: pronuclei; arrows: centrosomes. (J) Quantification of GFP::PAR-2 distributions in *air-1(RNAi)* embryos with sperm entry next to maternal pronucleus ("anterior") or opposite from it ("posterior"). (K) Embryos expressing GBP::mCh::AIR-1 (RNAi-resistant, denoted R) and mCh::PAR-2, depleted of endogenous AIR-1 either alone (left panel) or together with SPD-2 (middle panel). Right panel: hermaphrodites depleted of endogenous AIR-1 and SPD-2 were mated with GFP::SAS-7 males; see schematic for a description of the experimental set up. Note that GBP::mCh::AIR-1(R) is not cortical in embryos depleted of SPD-2. (L) Quantification of mCh::PAR-2 distributions corresponding to (K).

DOI: https://doi.org/10.7554/eLife.44552.022

The following figure supplement is available for figure 7:

**Figure supplement 1.** Embryos break symmetry in the absence of centrosomes.

DOI: https://doi.org/10.7554/eLife.44552.023

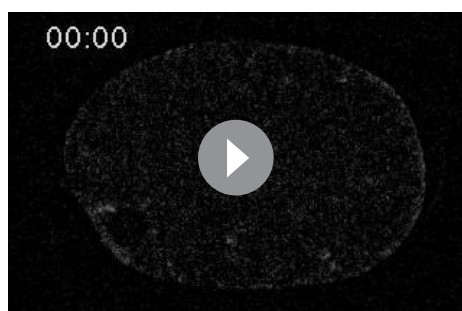

**Video 8.** Embryo expressing GFP::PAR-2 and GFP::SAS-7 that underwent successful laser ablation of the centrosome.
DOI: https://doi.org/10.7554/eLife.44552.024

## Discussion

Symmetry breaking is critical for organizing living matter. Our findings lead us to propose a novel mechanism for symmetry breaking during anterior-posterior axis specification in *C. elegans* (**Figure 8**). We propose that AIR-1 exerts a dual role in this process, acting early in the cell cycle, possibly in the cytoplasm, to prevent spontaneous polarization and at centrosomes thereafter to define the future posterior of the animal. In the absence of either AIR-1 or of centrosomes, embryos become bipolar, with PAR-2 domains at both poles. Our theoretical analysis reveals how this remarkable patterning defect can develop through centrosome-independent polarization, whereby the two poles of the embryo act as preferred sites of PAR-2 recruitment. Overall, our work reveals that the *C. elegans* zygote possesses the ability to spontaneously form polarity domains in a centrosome-independent manner at the onset of development.

## AIR-1 exerts a dual role in symmetry breaking

We discovered that AIR-1 localizes to the cell cortex prior to symmetry breaking, overlapping substantially with the cortical array of microtubules present at this early stage of development. We found also that SPD-2 and SPD-5 are needed for cortical AIR-1 distribution, in addition to their known requirement for centrosomal AIR-1 recruitment. In Xenopus egg extracts, interaction with the SPD-2 homologue CEP192 is required for autophosphorylation and thereby activation of Aurora A (*Joukov et al., 2014*), raising the possibility that SPD-2 is likewise needed for activating AIR-1 in *C. elegans*. This possibility is compatible with the observation that preventing AIR-1 activity using a kinase inactive mutant or a small molecule inhibitor results in cortical depletion of AIR-1 (*Kotak et al., 2016*). The dual requirement of SPD-2 and SPD-5 for correct AIR-1 distribution and/or activity both early and late in the cell cycle provides an explanation for why *spd-2*(*RNAi*) and *spd-5*(*RNAi*) embryos exhibit the same polarity defects as *air-1*(*RNAi*) embryos.

We found in addition that targeting excess AIR-1 to the vicinity of the plasma membrane is generally sufficient to prevent PAR-2 recruitment throughout the embryo, in a manner independent of SPD-2 and SPD-5. Therefore, cortical AIR-1 can act as a negative regulator of polarity establishment in some circumstances. However, when centrosomes are present, cortical AIR-1 is not needed to prevent polarity establishment, as evidenced by the fact that microtubule removal, which results in the loss of cortical AIR-1, does not impact A-P polarity. Furthermore, we found that endogenous cortical or cytoplasmic AIR-1 can limit the expansion of an anterior PAR-2 domain upon delayed meiotic cell cycle progression, and thus could be of importance under environmental stress conditions that cause such delays (*Davis-Roca et al., 2017*).

As the zygote progresses through the cell cycle, AIR-1 rapidly becomes enriched at centrosomes, in line with the fact that AIR-1 is very mobile (*Kress et al., 2013*). We found that targeting GBP::mCh::AIR-1 to centrioles in embryos depleted of SPD-2, and thereby of microtubule nucleation activity, is sufficient to recruit PAR-2 to the neighboring plasma membrane. Therefore, later in the cell cycle, centrosomal AIR-1 can act as a positive regulator of polarization.

What links the two facets of AIR-1 function in symmetry breaking? Our findings suggest two

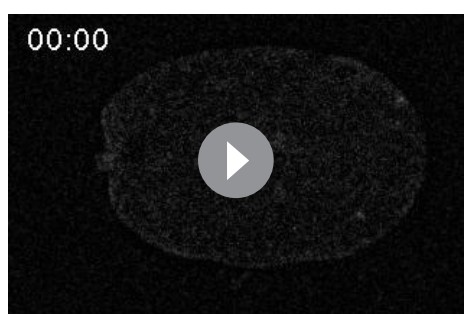

**Video 9.** Embryo expressing GFP::PAR-2 and GFP::SAS-7 that underwent unsuccessful laser ablation of the centrosome.
DOI: https://doi.org/10.7554/eLife.44552.025

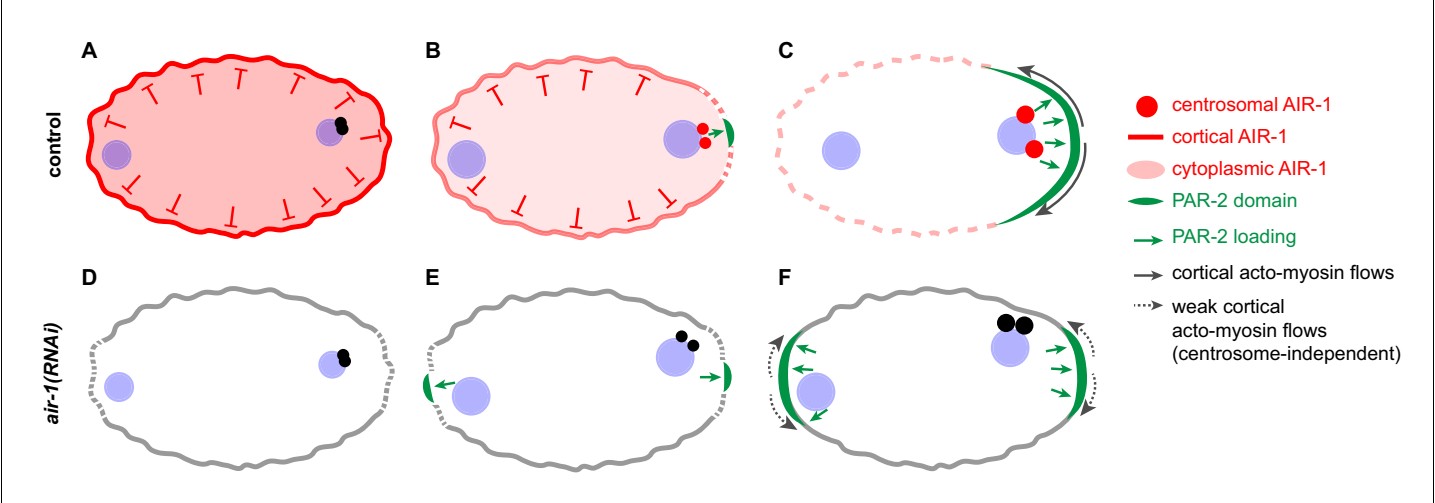

**Figure 8.** Working model of AIR-1 function during symmetry breaking in the wild type (**A–C**) and the bipolar phenotype in *air-1(RNAi)* embryos (**D–F**). (**A**) Prior to symmetry breaking in the wild type, AIR-1 localizes on the cell cortex and in the cytoplasm. (**B,C**) As the cell cycle progresses, AIR-1 is recruited onto centrosomes by SPD-2 and SPD-5, leading to AIR-1-mediated symmetry breaking in the vicinity. (**C**) Symmetry breaking is evidenced by the local recruitment of PAR-2 onto the plasma membrane and by anterior-directed cortical flows. (**D,E**) Upon *air-1(RNAi)*, symmetry is broken spontaneously at the two poles of the zygote, since AIR-1 can neither prevent spontaneous polarization early in the cell cycle nor promote a single symmetry breaking event later in the cell cycle. (**F**) Curvature-dependent PAR-2 membrane attachment increases the dissociation of NMY-2, thus creating a feedback loop that induces weak cortical flows toward the center from either side, which drives the recruitment of more PAR-2 and the segregation of anterior and posterior PAR proteins. See text for further information.

DOI: https://doi.org/10.7554/eLife.44552.026

possible scenarios. In both of them, AIR-1 acts initially throughout the embryo as a negative regulator of symmetry breaking, perhaps by phosphorylating a cortical substrate protein. Thereafter, in a first scenario, AIR-1 could undergo substrate switching upon relocalizing to centrosomes, and phosphorylate another substrate in that location, which would be critical for symmetry breaking, possibly through diffusion from the centrosomes to the neighboring cortex. In a second scenario, AIR-1 could have just one cortical substrate of relevance for polarity throughout the first cell cycle, in which case centrosomes would serve to locally deplete AIR-1, thereby ensuring uniqueness of symmetry breaking.

There are countless Aurora A kinase substrates from yeast to man (*Koch et al., 2011*; *Sardon et al., 2010*; *Tien et al., 2004*; *Willems et al., 2018*), and it will be interesting to identify the one(s) that serve to polarize the *C. elegans* zygote. Moreover, it will be valuable to explore whether the dual function uncovered in *C. elegans* extends to homologous proteins, including the human Aurora A oncogene.

## Centrosome-independent polarization of the *C. elegans* zygote

Centrosomes are sufficient to dictate symmetry breaking and thus polarity along the A-P axis of wild-type *C. elegans* zygotes. Indeed, the position of centrosomes within the embryo determines the location of symmetry breaking and of PAR-2 recruitment to the plasma membrane (*Bienkowska and Cowan, 2012*; *Cowan and Hyman, 2004*; *Cuenca et al., 2003*; *Goldstein and Hird, 1996*). Furthermore, centrosomes have been proposed to be necessary for A-P polarity, since their ablation by a laser microbeam has been reported to result in the absence of PAR-2 throughout the embryo (*Cowan and Hyman, 2004*). In contrast, using a centriolar marker bright enough to conduct physical centrosome ablation experiments at early time points, we found here that embryos usually recruit PAR-2 in a bipolar fashion in the absence of centrioles, as they do in *air-1(RNAi)* embryos. The timing of centrosome ablation might be critical for the outcome of such experiments and potentially might explain the discrepancy between our results and the earlier work. Since symmetry breaking as evidenced by cortical flows occurs before the posterior PAR-2 domain becomes detectable (see *Figure 4—figure supplement 1C*), it is plausible that centrosome ablation in the

earlier study occurred when symmetry breaking had been initiated. Perhaps such initiation would no longer allow for spontaneous symmetry breaking and thus result in the complete absence of cortical PAR-2. Regardless, together with the genetic centrosome ablation experiments conducted with *such-1* mutant sperm, our findings from the physical centrosome ablations firmly establish that polarization, albeit aberrant, occurs without centrosomes in *C. elegans*.

The mechanism that governs spontaneous polarization of the zygote in the absence of AIR-1 or of centrosomes is largely independent of cortical flows. Indeed, two PAR-2 domains can form in *nmy-2(ne3409) air-1(RNAi)* embryos, where cortical flows are essentially absent or below the detection limit. However, PAR-2 domains in such embryos are significantly smaller and develop later than in *air-1(RNAi)* embryos. Our findings also establish that the mechanism leading to spontaneous polarization in the absence of AIR-1 or of centrosomes functions independently of microtubules. Given this, it is intriguing that GFP::PAR-2 R163A or R183-5A, two mutants with impaired microtubule binding, fail to recruit PAR-2 onto the plasma membrane in *air-1(RNAi)* embryos. A potential explanation for this conundrum is that these mutants are impaired in more functions than just microtubule binding, for instance in PAR-2 oligomerization (*Arata et al., 2016*). In conclusion, our findings indicate that polarized membrane domains can be initiated largely independently of cortical flows through the instability of the PAR system, but that cortical flows help sharpen and accelerate polarization of the embryo.

## A physical basis for centrosome-independent polarization mechanisms

We analyzed the dependence between PAR proteins and cortical flows in polarizing the zygote by coupling a reaction-diffusion-advection system to active gel theory. In previous work, cortical flows were considered to be induced by centrosomes, which cannot act as a trigger in *air-1(RNAi)* embryos, nor following their genetic or physical ablation. Instead, we assumed that PAR-2 indirectly reduces cortical contractility via antagonism with PAR-6 (*Gross et al., 2019*). Thereby, PAR-2 distribution can result in gradients of contractility and thus of cortical flows despite the absence of centrosomes.

Based on the findings with *air-1(RNAi)* embryos, including when placed into triangular PDMS chambers, we introduced a preferential attachment rate $\alpha$ of PAR-2 in the polar regions of the embryo in the dynamical system. Although we demonstrated experimentally that PAR-2 domains form preferentially in such regions, the underlying mechanism remains to be clarified further. Perhaps the preference stems from distinct lipid distributions in curved regions that could influence the attachment rate of PAR-2, which can bind phospholipids (*Motegi et al., 2011*). Moreover, purely geometric effects can cause preferential attachment at the poles (*Rangamani et al., 2013*) and have been suggested to play a role for PAR protein localization in *C. elegans* (*Gessele et al., 2018*). However, whereas geometry has been shown to be important for some systems of protein self-organization (*Wettmann and Kruse, 2018*), how exactly it contributes to PAR-2 membrane recruitment remains to be deciphered.

Regardless, the physical description we developed successfully captures the phenotypes of wild-type embryos, of embryos depleted of AIR-1, as well as of embryos placed in triangular chambers. It will be of interest to investigate embryos placed in differently shaped PDMS chambers, as this might help uncover the mechanism enabling curvature preference and further develop the physical model. Moreover, we realize that a potential limitation of our current physical analysis stems from having been performed in 1D. A future analysis using a 2D geometry will likely uncover further aspects of the self-polarization mechanism, including by explicitly accounting for the impact of centrosomes and geometry, as well as by providing a full description of cortical dynamics.

## Self-organized polarity establishment across eukaryotes

Spontaneous polarization mechanisms such as the one revealed here for *C. elegans* are encountered in other systems. For instance, a self-enhancing positive feedback loop of Cdc42-GTP contributes to polarization of the emerging bud in *S. cerevisiae* (*Bendezú et al., 2015*). Another example occurs in HeLa tissue culture cells forced to undergo monopolar division, which exhibit spontaneous symmetry breaking manifested by a concentration of actin, microtubules and cleavage furrow components on one side of the cell (*Hu et al., 2008*). Spontaneous symmetry breaking can also be recapitulated in cell-free assays, including in actin cortices formed at the periphery of Xenopus egg extracts

encapsulated in an oil emulsion (*Abu Shah and Keren, 2014*). Likewise, beads coated with *Listeria monocytogenes* ActA, which catalyzes actin polymerization, spontaneously break symmetry and display directional motion due to small asymmetries of the polymerized actin around the bead (*van Oudenaarden and Theriot, 1999*). Our work echoes such findings from other domains of life and demonstrates that spontaneous polarization can also occur during the very first step of development in a metazoan organism. Such a centrosome-independent symmetry breaking mechanism could be of particular importance in systems devoid of centrioles such as parthenogenetic species. In the wild-type *C. elegans* zygote, AIR-1 funnels the inherent self-organizing properties of the cell onto a single site, next to paternally contributed centrioles, thus ensuring spatial coupling between paternal and maternal contributions at the onset of development.

## Materials and methods

### Worm strains

Nematodes were maintained at 24°C using standard protocols (*Brenner, 1974*). Worms carrying temperature-sensitive mutations were maintained at the permissive temperature (16°C) and shifted to the restrictive temperature (24°C) for 24 hr prior to imaging. See *Supplementary file 1* for the list of worm strains used in this study. New transgenic worm strains were generated as described just hereafter:

The GBP::mCherry::AIR-1 construct was obtained by inserting the mCherry coding sequence at the 3'-end of the GFP-binding protein (GBP) coding sequence, obtained from the pAOD-VHHGFP4 vector (a kind gift from Aurelien Olichon), followed by codon optimized RNAi-resistant *air-1* cDNA (a kind gift from Asako Sugimoto). Flexible linkers encoding thirteen amino acids (GAGAGAGAGAFSV) were inserted between GBP and mCherry cDNAs, as well as between mCherry and AIR-1 cDNAs. The corresponding integrated transgenic line was generated by microparticle bombardment of *unc-119(ed3)* mutant worms (*Praitis et al., 2001*).

CRISPR/Cas9-mediated genome editing was used as described previously (*Dickinson et al., 2015*) to insert tagRFP-T in frame at the 5' of the *sas-7* locus. To target Cas9 to the *sas-7* locus, the 5'gcttaaaatcaactcaccg(TGG)3' −19 bp targeting sequence was inserted into the Cas9-sgRNA construct (pDD162) (Addgene). Homologous repair template for insertion of the self-excising selection cassette (SEC) was generated by modifying the pDD284 (Addgene) vector using Gibson assembly. Left and right homology arms of 655 bp and 652 bp were PCR-amplified, before insertion into pDD284 (Addgene) opened using ClaI, SpeI and PsiI to remove the Flag tag from the original vector. A silent mutation in the PAM motif (5'TGG3' –>5'TAG') was introduced to prevent repair template cutting and a 5'CACTCCACTGGAACCTCTAGA3'−21 bp linker was added between the *tagRFP* and *sas-7* coding sequences. An injection mix containing 50 ng/µl targeting vector, 50 ng/µl homologous repair template, 10 ng/µl pGH8 (*Prab-3::mcherry::unc-54 3'UTR*) (Addgene), 5 ng/µl pCFJ104 (Addgene) (*Pmyo-3::mcherry::unc-54 3'UTR*) and 2.5 ng/µl pCFJ90 (*Pmyo-2::mcherry::unc-54 3'UTR*) (Addgene) was micro-injected into the gonads of wild type (N2) worms. Selection of insertion events and removal of the SEC cassette was performed as described (*Dickinson et al., 2015*).

A similar strategy was used to tag endogenous SPD-2 with RFP. The 5'tgttcattacagagattcat(TGG) 3'−21 bp targeting sequence was cloned into the pDD162 (Addgene) vector and pDD286 (Addgene) was modified using Gibson assembly and Q5-directed mutagenesis to generate the homologous repair template. PCR amplified 639bp-left and 667bp-right homology arms were inserted into pDD286 (Addgene) digested with ClaI and NgoMIV; six silent mutations were introduced in the targeted sequence and the PAM site 5'tgctcgttgcacaggttcat(GGG)'3. As described above, an appropriate injection mix was micro-injected in the gonads of wild type N2 worms; screening was performed as described (*Dickinson et al., 2015*).

### RNAi

RNAi-mediated depletion was performed essentially as described (*Kamath et al., 2001*), using bacterial feeding strains from the Ahringer (*Kamath and Ahringer, 2003*) or Vidal libraries (*Rual et al., 2004*) (the latter a gift from Jean-François Rual and Marc Vidal). The bacterial feeding strain for depleting specifically endogenous AIR-1 without affecting *air-1* RNAi resistant transgene (*tjls173* and *tjls188*) expression was a kind gift from Asako Sugimoto (*Toya et al., 2011*). The bacterial feeding

strains used to deplete specifically endogenous PAR-2 in strains expressing RNAi-resistant *par-2* transgenes (*axIs1933*, *axIs1936*, *axIs1934*) was a kind gift from Fumio Motegi and Geraldine Seydoux (*Motegi et al., 2011*).

RNAi for *air-1* (Vidal), *rga-3* (Vidal), *spd-2* (Vidal), *spd-5* (Vidal) or *par-2* (Ahringer) was performed by feeding L3-L4 animals with bacteria expressing the corresponding dsRNA at 24°C for 16–26 hr. In *Figure 1—figure supplement 1B*, *air-1(RNAi)* was diluted 1:5 or 1:10 with bacteria expressing an empty vector (L4440). RNAi for *tbg-1* (Ahringer) and *tbg-1 +air-1* was performed by feeding L2-L3 animals with bacteria expressing dsRNAs at 20°C for 48 hr and imaging their embryos. RNAi for *tba-2* (Ahringer) and *tba-2 +air-1* was performed by feeding L3 animals with bacteria expressing dsRNA at 24°C for 30–36 hr and imaging their embryos. Strains carrying temperature sensitive mutations were shifted to the restrictive temperature (24°C) for 24 hr and RNAi performed as described above.

## Live imaging

Gravid hermaphrodites were dissected in osmotically balanced blastomere culture medium (*Shelton and Bowerman, 1996*) and the extracted embryos mounted on a 2% agarose pad. Dual fluorescence and DIC time lapse microscopy was performed at room temperature with a 60x CFI Plan Apochromat Lambda (NA 1.4) objective on a Nikon Eclipse Ti-U Inverted Microscope connected to an Andor Zyla 4.2 sCMOS camera, or with a 63x Plan-Apochromat (NA 1.4) objective on a Zeiss ObserverD.1 inverted microscope connected to the same type of camera. One frame was captured every 10 s, and a z-stack was acquired at every time point, covering 20 μm, with a distance of 0.7 μm between focal planes. Time lapse images in *Figure 5A,C,J,K*; *Figure 3—figure supplement 1E*; *Figure 5—figure supplement 1B* were acquired at 23°C using an inverted Olympus IX 81 microscope equipped with a Yokogawa spinning disk CSU - W1 with a 63x (NA 1.42 U PLAN S APO) objective and a 16-bit PCO Edge sCMOS camera. Images were obtained using a 488-nm solid-state laser at 60% laser power, with an exposure time of 400 ms. Embryos were imaged every 5 s, and a z-stack was acquired at every time point, covering 2 μm of the embryo cortex, with a distance of 0.5 μm between focal planes (i.e. four cortical planes).

## Nocodazole treatment

Worms were dissected in 10 μg/ml Nocodazole (Sigma Aldrich, M1404), diluted in 3.3% DMSO and 96.7% osmotically balanced blastomere culture medium (*Shelton and Bowerman, 1996*); the extracted embryos were mounted on a 2% agarose pad. The drug entered the embryos due to the permeability of the eggshell during meiosis, and was effectivee in this manner, as evidenced notably by the lack of pronuclear migration. Epifluorescence imaging was performed as described above.

## Centrosome laser ablation

We conducted laser ablation on a previously described set-up (*Mayer et al., 2010*). Ablation was performed by applying 10 ultraviolet pulses at 1 kHz at equidistant sites on the circumference of a circle 0.5 μm in diameter. The centrosome was ablated at the end of meiosis II, before the centrioles moved towards the cortex. After ablation, the zygote was imaged every 10 s until nuclear envelope breakdown. Embryos were monitored for the re-appearance of the centrosomal signal, which occurred when the centrosome was only bleached and not destroyed, as well as for cell rupture or for membrane damage, which resulted in cytoplasmic leakage; all such embryos were excluded from the analysis.

## Image processing and analysis

Images acquired at the spinning disk were processed by z-projecting the four cortical planes using maximum intensity (ImageJ software). Images acquired with the epifluoresence microscope (Zeiss ObserverD.1or Nikon Eclipse Ti-U) were processed as follows using ImageJ: images underwent background subtraction using a 'rolling ball' algorithm of 10 pixels, followed by a maximum intensity z-projection. Grey levels were set identically for all images within each experimental series.

For flow velocity analysis, heat maps were obtained using Particle Image Velocimetry (PIV) with a freely available PIVlab MATLAB algorithm (pivlab.blogspot.de). Using PIVlab, we performed a four-step multi pass with a final interrogation area of 8 pixels with a step of 4 pixels. 2D velocity fields were obtained by averaging the x-component of velocity along the y-axis for each value in a single

frame. All values were averaged over several embryos and represented in a heat map using MAT-LAB. Prior to averaging all values across multiple embryos, embryos were aligned temporally using the best fit of the pronuclei growth curves (*De Simone et al., 2016*). Pronuclear diameter was measured from the DIC images using ImageJ. For *Figure 7—figure supplement 1A*, centrosomes were tracked in three dimensions using the ImageJ plugin TrackMate (imagej.net/TrackMate).

## PAR-2 domain extent and intensity measurements

Images were processed as described above using ImageJ following straightening of the embryo's outline for each time point using the ImageJ plugin 'Straighten' (imagej.nih.gov/ij/plugins/straighten.html). The intensity of each pixel along the embryo's straightened circumference was measured with ImageJ, and the resulting values displayed using Matlab in a heat map representing the GFP::PAR-2 intensity.

To calculate the extent of the PAR-2 domain, the circumference of the cell was smoothened in Matlab for each time point using a 23-point moving average to reduce acquisition noise. For each time point, a noise floor estimation was obtained by detecting the max and min values following a smoothing by applying a moving average. The noise floor value was then approximated using the minimum values. The value of the average background noise was used as a threshold to determine the points constituting the domain boundary. The same threshold was applied to all analyzed image sequences. In order to reduce noise-induced threshold crossing, the data values were squared.

## Mating experiments

SUCH-1 depletion causes, in addition to the paternal sperm phenotype, a maternally contributed delay in mitosis. We used *such-1(t1168)* males to fertilize *fem-1(hc17)* females, thus ensuring that all embryos resulted from *such-1(t1168)* mutant sperm and that the maternal contribution was normal. Gravid *fem-1(hc-17)* hermaphrodites expressing RFP::NMY-2; GFP::PAR-2; GFP::SAS-7 were shifted to the restrictive temperature (24°C) and L4 progeny were then mated with *such-1(t1668)* males at 20°C for 24 hr. Gravid adults were dissected and embryos imaged as described above. Prior to and after imaging, each embryo was thoroughly screened for the absence of centrosomes as evaluated by the lack of focal GFP::SAS-7 signal.

To target GBP::mCherry::AIR-1 to centrosomes, L4 larvae expressing mCherry::PAR-2 and GBP::mCherry::AIR-1 were mated with GFP::SAS-7 males for 24 hr at 20°C on *spd-2(RNAi)*. Gravid adults were dissected and embryos imaged as described above. Only embryos that were fertilized, as evaluated by the presence of GFP::SAS-7, were imaged.

## Immunofluorescence

Embryos were permeabilized by freeze-cracking, followed by fixation in methanol at −20°C for 5 min and by incubation for 1 hr at room temperature with the following primary antibodies: mouse anti-α-tubulin, 1/200, (DM1A, Sigma), rabbit anti-PAR-2, 1/200 (*Pichler et al., 2000*), rabbit anti-AIR-1, 1/1000 (*Hannak et al., 2001*), chicken anti-GFP, 1/200 (Abcam). Secondary antibodies were Alexa-Fluor-647-coupled anti-mouse-IgG, Alexa-Fluor-488-coupled anti-chicken-IgY and Alexa-Fluor-568-coupled anti-rabbit-IgG, all used at 1:500 (Jacskon laboratories). Slides were counterstained with 1 mg/ml Hoechst 33258 (Sigma) to reveal DNA.

## PDMS chambers

Micro-well structures were produced using standard soft-lithography methods. For triangular PDMS chambers, a custom lithography mask of a silicon waver coated with AZ 15nXT (MicroChemicals GmbH, Ulm, Germany) was illuminated with UV to generate a casting mould of triangles with a side length of ~40 µm. Micro-well structures were generated by pouring poly(dimethylsiloxane) (PDMS; Sylgard 184, VWR) into these casting moulds. Finally, triangular PDMS structures were cured for 1.5 hr at 75°C. Before each experiment, PDMS chambers were treated with oxygen plasma for 90 s to reduce hydrophobicity. Worms were dissected as described above, and embryos placed into the triangular chamber with an eyelash tool prior to symmetry breaking. Epifluoresence imaging was performed as described above.

PDMS cylinders with a diameter of 35 µm utilized in *Figure 3—figure supplement 1D* were generated in an analogous manner, and embryos also placed into the cylindrical chamber with an

eyelash tool prior to symmetry breaking. Embryos entered the chamber with the anterior or posterior pole first, allowing imaging of one of the other pole using spinning disk confocal microscopy as described above.

## Statistical analysis

Statistical significance for all shown quantification, except for *Figure 3E*; *Figure 6—figure supplement 1B,C*; *Figure 7—figure supplement 1D*, was tested using the non-parametric two-sided Fisher's exact test, comparing a single category against the other pooled categories (e.g. posterior vs non-posterior) for two conditions (e.g. control versus air-1(*RNAi*)). The corresponding p-values for all experiments can be found in *Supplementary file 2*. A p-value<0.05 was considered as statistically significant. Statistical significance in *Figure 3E*; *Figure 6—figure supplement 1B,C* and *Figure 7—figure supplement 1D* was tested using the non-parametric Mann-Whitney-U test. A p-value<0.05 was considered as statistically significant.

## Acknowledgements

We are grateful to Carrie Cowan and Sachin Kotak for sharing unpublished observations and for fruitful discussions. For their gift of worm strain to Bruce Bowerman, Fumio Motegi, Asako Sugimoto, Jessica Feldman, Anthony Hyman and Geraldine Seydouxs, as well as the *Caenorhabditis* Genetics Center (CGC), which is funded by NIH Office of Research Infrastructure Programs (P40 OD010440). We thank Alexandra Bezler, Alessandro De Simone and Radek Jankele for comments on the manuscript. We thank the Microstructure Facility (CMCB, TU-Dresden), in part funded by the State of Saxony and the European Fund for Regional Development — EFRE, for the production of the triangular micro-well chambers. This work was supported by post-doctoral fellowships from EMBO to KeKl (ALTF 81–2017) and to MP (ALTF 1426–2016), the Fondation Bettencourt.Schueller prize to NL, as well as grants from the Swiss National Science Foundation to PiGö (31003A_155942) and to KaKr, (205321_175996), and from the European Research Council (281903 and 742712) to SWG.

## Additional information

### Funding

| Funder | Grant reference number | Author |
|---|---|---|
| European Molecular Biology Organization | ALTF 81-2017 | Kerstin Klinkert |
| Fondation Bettencourt Schueller | | Nicolas Levernier |
| European Molecular Biology Organization | ALTF 1426-2016 | Marie Pierron |
| H2020 European Research Council | 281903 | Stephan W Grill |
| H2020 European Research Council | 742712 | Stephan W Grill |
| Deutsche Forschungsgemeinschaft (DFG) under Germany ́s Excellence Strategy | EXC-2068 – 390729961 | Stephan W Grill |
| Swiss National Science Foundation | 205321_175996 | Karsten Kruse |
| Swiss National Science Foundation | 31003A_155942 | Pierre Gönczy |

The funders had no role in study design, data collection and interpretation, or the decision to submit the work for publication.

## Author contributions
Kerstin Klinkert, Conceptualization, Data curation, Software, Formal analysis, Validation, Investigation, Visualization, Methodology, Writing—original draft, Project administration, Designed the project, Conducted most experiments with support from LvT, SH and CB, Analyzed the data; Nicolas Levernier, Data curation, Software, Formal analysis, Investigation, Visualization, Methodology, Writing—original draft, Developed the physical description; Peter Gross, Stephan W Grill, Resources, Writing—review and editing, Developed the PDMS chambers; Christian Gentili, Investigation, Generated worm strains; Lukas von Tobel, Sarah Herrman, Investigation, KeKl conducted most experiments with support from LvT, SH and CB; Marie Pierron, Investigation, Writing—review and editing, Generated worm strains; Coralie Busso, Investigation, KeKl Conducted most experiments with support from LvT, SH and CBl; Karsten Kruse, Conceptualization, Resources, Supervision, Funding acquisition, Writing—original draft, Developed the physical description; Pierre Gönczy, Conceptualization, Resources, Supervision, Funding acquisition, Writing—original draft, Designed the project, Analyzed the data

## Author ORCIDs
Kerstin Klinkert (iD) https://orcid.org/0000-0001-5268-3710
Pierre Gönczy (iD) https://orcid.org/0000-0002-6305-6883

## Decision letter and Author response
Decision letter https://doi.org/10.7554/eLife.44552.033
Author response https://doi.org/10.7554/eLife.44552.034

## Additional files

### Supplementary files
• Supplementary file 1. List of worm strains.
DOI: https://doi.org/10.7554/eLife.44552.027

• Supplementary file 2. Statistical analysis of all bar graphs.
DOI: https://doi.org/10.7554/eLife.44552.028

• Transparent reporting form
DOI: https://doi.org/10.7554/eLife.44552.029

### Data availability
All data is available in the manuscript or the supplementary materials.

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

## Appendix 1

DOI: https://doi.org/10.7554/eLife.44552.030

# Theoretical description of symmetry breaking with and without external cue

## A) Description of the framework

The process of polarization depends on competitive binding of anterior and posterior proteins to the plasma membrane coupled to cortical actin flows. These processes can be described by a non-linear advection-reaction-diffusion system (*Goehring et al., 2011*). This can successfully describe polarization of wild-type embryos when an appropriate cortical actin flow is imposed, reflecting the centrosome-derived cue. The dynamic equations in this system for the concentrations $A$ and $P$ of membrane-bound anterior (PAR-6) and posterior (PAR-2) proteins, respectively, are

$$\partial_t A = D_A \partial_x^2 A - \partial_x(vA) + k_{on,A}A_{cyto} - k_{off,A}A - k_{AP}PA \tag{1}$$

$$\partial_t P = D_P \partial_x^2 P - \partial_x(vP) + k_{on,P}P_{cyto} - k_{off,P}P - k_{PA}A^2P. \tag{2}$$

Here, $D_{A,P}$ denote the diffusion coefficients of the two species, $v$ the velocity of the cortical flow, $k_{on}$ and $k_{off}$ the respective attachment and detachment rates, and $A_{cyto}$ and $P_{cyto}$ their cytoplasmic concentrations. The latter are equal to the total number of proteins minus the membrane bound proteins divided by the embryo volume. The last terms $k_{AP}PA$ and $k_{PA}A^2P$ account for the mutual antagonism between anterior and posterior PAR proteins (*Goehring et al., 2011*). In the equations, we exploit that the embryo is symmetric with respect to rotations around the long axis and describe the concentrations only as a function of the remaining coordinate along the membrane, which is denoted by $x$. We scale lengths, such that $0 \leq x < 2\pi$. The equations are complemented by periodic boundary conditions. Then, cytoplasmic and membrane concentrations of the anterior proteins are related through $A_{cyto} = \rho_A - \psi \int_0^{2\pi} A(x)dx$ where $\rho_A$ is the total amount of proteins divided by the volume of the embryo (i.e. cytoplasmic concentration if there is no protein on the membrane), and $\psi$ the area/volume ratio of the embryo.

The reaction kinetics can exhibit bi-stability, such that for a specific set of parameter values either an $A$-dominant state with high $A$ and low $P$ or a $P$-dominant state with high $P$ and low $A$ is reached. There is also an unstable state with intermediate concentration of both $A$ and $P$. These homogenous, unpolarized states give way to a polarized state if a strong enough flow is applied, with one part of the embryo being in the $A$-dominant state (anterior domain) and the other in the $P$-dominant state (posterior domain). Once the polarized state is established, it persists even when the flow ceases (*Goehring et al., 2011*).

A constrained set of parameters for which the above behavior holds has been determined experimentally (*Goehring et al., 2011*). Here, we use similar values (see Table in C, below) with one important modification: our experimental observations suggest that the attachment rate of the posterior proteins is higher in regions with larger curvature. We account for this observation by imposing $k_{on,P}(x) = k^0[1 + \alpha(C(x) - C_{min})]$, where $C$ is the membrane curvature at position $x$. Furthermore, $k^0$ is the rate of attachment to the minimally curved part of the embryo and $\alpha$ quantifies the strength of the coupling between the curvature and the PAR-2 attachment rate. For $k^0$ we take the value obtained by *Goehring et al. (2011)*.

The velocity $v$ of the cortical actin flow remains to be specified in this novel framework. In most previous work, flows were assumed to be directly linked to the action of centrosomes at the posterior pole and imposed *ad hoc*, even though the exact nature of this action and its temporal regulation remain unclear (*Goehring et al., 2011*; *O'Neill et al., 2009*; *Tostevin and Howard, 2008*). We complement the above equations with an explicit description of actin

dynamics, which will notably yield an actin flow. Similar to Ref. (*Joanny et al., 2013*), we base our description on actin conservation and force balance. Explicitly,

$$\partial_t \rho = -\partial_x(v\rho) + D\partial_x^2\rho \tag{3}$$

$$\eta\partial_x^2 v = \partial_x\Pi + \gamma v, \tag{4}$$

where $\rho$ is the cortical acto-myosin gel density, $D$ an effective diffusion constant, $\eta$ its viscosity, and $\gamma$ quantifies the gel friction against the membrane. The function $\Pi$ denotes an effective cortical stress and, in particular, accounts for NYM-2-induced contractility. Gradients in $\Pi$ along the membrane generate flows over a typical distance $l = \sqrt{\eta/\gamma}$. Following Ref. (*Joanny et al., 2013*), we choose $\Pi = -a\rho^3 + b\rho^4$ with $a$ and $b>0$ so that the cortex is contractile for moderate densities and that the common positive pressure is recovered for high densities. The experiments suggest that PAR-2 loading affects actin mechanics; in particular, there are no flows in the absence of PAR-2 and AIR-1 (*Figure 2—figure supplement 1N*). We assume that cortex contractility decreases with increasing PAR-2 concentration and account for this by setting $a = a^0 - \beta P$. In contrast, we choose the passive coefficient $b$ to be uniform. In our phenomenological description, we do not specify a molecular mechanism through which $P$ reduces contractility, as this is irrelevant for the qualitative behavior of the system. Previous work suggests that posterior PAR proteins decrease the concentration of active myosin through the RHO-1 pathway. However, since we focus here on the essential parts of the physical mechanism underlying symmetry breaking, we chose not to include these molecular details. Likewise, we do not consider the actin network and the myosin distributions separately, even though this distinction might be relevant in other situations.

Similar to *Equation (1) and (2)*, the full set of *Equation (1) - (4)* has an $A$-dominant and a $P$-dominant stationary state. Due to the dependence of $P$ binding on the curvature, however, they are not completely homogenous. For a sufficiently strong preferential attachment rate $\alpha$ or decrease of contractility $\beta$, these states are not stable: $P$ will bind preferentially to the membrane in a curved region, which generates a small gradient of $P$ along the membrane. This protein gradient leads to a gradient of contractility, which in turn generates an actin flow directed away from it.

The wild-type situation is obtained by locally weakening the cortex at the posterior pole, mimicking the centrosomal cue, which induces cortical flow from the posterior pole to the anterior pole. This is achieved by initially disrupting the cortex (that is setting $\rho = 0$) at the posterior pole. We get qualitatively similar results by reducing activity (e.g. setting $a = 0$) at this pole. Then, embryos always develop a posterior PAR-2 domain, similar to what was described previously (*Goehring et al., 2011*), where the flow had been inserted manually. Strikingly, our model constrains the PAR domains to lie at the poles of the embryo, even if the cue is not delivered on a pole. Moreover, if we shift all densities of the polarized state by at most $\pm\pi/2$ along $x$, the system spontaneously returns to its original state, offering a way to understand the robust positioning of posterior PAR proteins at the cell poles.

### Typical *air-1(RNAi)* embryo

We first consider a typical embryo of length 57 μm (*Goehring et al., 2011*) and aspect ratio 1.6 without centrosomal cue. Results of varying $\alpha$ and $\beta$ are given in *Figure 3I* . For small $\alpha$ or small $\beta$, the embryo never polarizes and stays in an $A$-dominant state. Increasing $\alpha$ for a given $\beta$, we get first polar embryos (anterior or posterior). Increasing $\alpha$ further leads to coexistence of phases where either polar or bipolar embryos are obtained, depending on the initial random condition. Finally, increasing $\alpha$ further still leads to 100% of bipolar embryos.

### Triangles

We then apply our description to embryos squeezed in triangular chambers. Contour length is now approximately 120 μm, and the area/volume ratio is $0.22\mu m^{-1}$. We model the curvature function as $C(x) = C_0\left[1 - 0.6\cos(3\pi x/L)^2\right]^{-6}$ with $C_0 = 0.05\mu m^{-1}$. This gives rise to phenotypes

with one, two or three domains, depending on the parameters $\alpha$ and $\beta$. Due to increased PAR-2 domain extension in this triangular shape, the bipolar phenotype appears to be unstable, leading to a shift towards a single PAR-2 domain.

## B) Parameters

Parameters for the advection-reaction-diffusion system are chosen following Goehring et al. (**Goehring et al., 2011**). The (unknown) rates of mutual antagonism $k_{AP}$ and $k_{PA}$ are chosen such that the system is in the bi-stable phase where $A$-dominant and polarised states coexist and are stable (phase (ii) (**Goehring et al., 2011**)).

The hydrodynamic length of the actin cortex is set to 15 µm, following Ref. (**Mayer et al., 2010**). The effective diffusion constant $D$ is obtained by studying experimental results for the steady-state actin density in a wild type-polarized state, balancing advective and diffusing flux, yielding $D \simeq 0.3 \mu\mathrm{m}^2/s$. The associated diffusive time scale is thus $\tau = l^2/D = 600s$. Finally, parameters $a$ and $b$ remain unknown but do not affect qualitatively the resulting phenotypes. We chose a set such that a uniform actin cortex is stable, and that the typical velocity magnitude of cortical flows is 4 µm/min. Actin density $\rho$ is normalized to unity and PAR-2 density for normal PAR-2 expression; $\rho_P^0$ is also set to unity.

| Parameter | Values |
|---|---|
| $\rho_A$ | $1.8 \mu m^{-3}$ |
| $\rho_P$ | $1 \mu m^{-3}$ |
| $D_A$ | $0.3 \mu m^2 s^{-1}$ |
| $D_P$ | $0.15 \mu m^2 s^{-1}$ |
| $k_{on,A}$ | $8.6 \times 10^{-3} \mu m s^{-1}$ |
| $k_{off,A}$ | $5.4 \times 10^{-3} s^{-1}$ |
| $k^0$ | $4.7 \times 10^{-2} \mu m s^{-1}$ |
| $k_{off,P}$ | $7.3 \times 10^{-3} s^{-1}$ |
| $k_{AP}$ | $0.19 \mu m^2 s^{-1}$ |
| $k_{PA}$ | $2.6 \mu m^4 s^{-1}$ |
| $D$ | $0.3 \mu m^2 s^{-1}$ |
| $l = \sqrt{\eta/\gamma}$ | $15 \mu m$ |
| $a^0$ | $90 \mu m^{-3} s^{-1}$ |
| $b$ | $-68 \mu m^{-4} s^{-1}$ |

## C) Simulations

**Equations (1)-(4)** are solved numerically using a finite-difference scheme. In each time-step, cortical velocity is obtained in Fourier space by solving (4) and back-transforming to real space. Then, actin density and protein densities are updated in Fourier space using Fourier-transform of (1-3), and then back-transformed to real space. The spatial discretization step is 0.7 µm.

Unless specified, the initial state of A- and P-proteins is $A_0(1 + 0.025\eta_1(x))$ and $P_0(1 + 0.025\eta_2(x))$ where $(A_0, P_0)$ is the $A$-dominant state, and $\eta_1(x)$ and $\eta_2(x)$ are independent and identically distributed random variable in [-1, 1]. All simulation kymographs were obtained with $\alpha = 1.2$ and $\beta = 100$.

