## [Decision Letter]

Thank you for submitting your article "Aurora A depletion reveals centrosome-independent polarization mechanism in *C. elegans*" for consideration by *eLife*. Your article has been reviewed by three peer reviewers, including Yukiko M Yamashita as the Reviewing Editor and Reviewer #1, and the evaluation has been overseen by Anna Akhmanova as the Senior Editor.

The reviewers have discussed the reviews with one another and the Reviewing Editor has drafted this decision to help you prepare a revised submission.

This is a very elegant and careful study that makes a significant contribution to understanding symmetry breaking in the *C. elegans* embryo. While it has been known for over 10 years that the centrosome is a key player in breaking symmetry to establish PAR polarity domains, the molecular mechanism has been elusive. This study by Klinkert et al., gives significant new insight by identifying a central role for the conserved Aurora kinase A (AIR-1). Their results lead to the model that AIR-1 inhibits actomyosin flow and symmetry breaking at the cortex early in the cell cycle, but then later AIR-1 is positively required at the centrosome to induce a posterior (PAR-2 loading) domain. The authors also show that in the total absence of AIR-1 (i.e. lack of inhibition), symmetry is broken spontaneously by loading of PAR-2 at the cortex, apparently due to regions of high membrane curvature. They also use microfabricated chambers to show that PAR-2 preferentially localizes to highly curved membranes in the absence of a centriolar cue, and that this preference explains the preference of PAR-2 for the poles in the absence of the normal centriolar/AIR-1-dependent cue that promotes polarity. The authors use mathematical modeling to provide further support for such a model. These are very interesting and impressive experiments and results that provide novel insight into the mechanisms that regulate and establish A-P polarity in the *C. elegans* zygote, a premiere model for understanding the establishment of cell polarity more generally. Overall the study is well done, with live imaging, thorough quantification, and some very nice technical advances for modifying cell shape and targeting proteins to specific locations. Overall, the results are convincing, and the paper is well organized and easy to follow (with a few exceptions).

The reviewers noted a few concerns that must be addressed prior to publication.

Essential revisions that require additional experiments:

1) In Figure 5, the authors show that cortical AIR-1 co-localizes to a substantial extent with cortical microtubules (MTs), and that AIR-1 is not required for cortical MTs (if anything, cortical MTs are increased in the absence of AIR-1). However, the authors do not address whether cortical AIR-1 requires microtubules. They show earlier in the paper that the bipolarity phenotype observed after AIR-1 knockdown does not depend on MTs (even though it does depend on PAR-2 constructs that can bind MTs, and they address this paradox well in the Discussion). Furthermore, it is known that polarity establishment does not require MTs, as the authors point out. But given the authors' convincing documentation of AIR-1 and cortical microtubule co-localization, and the authors' use of nocodazole to examine the polarity phenotypes in the absence of microtubules and AIR-1, it is a very easy experiment to address the rather obvious question as to whether cortical AIR-1 requires microtubules: is AIR-1 still enriched at the cortex after strong nocodazole treatment? This is an important experiment: if AIR-1 does require microtubules to be at the cortex, then one would expect nocodazole treatment to phenocopy AIR-1 knockdown, and/or one would have to consider the possibility that AIR-1 does not need to be at the cortex to inhibit PAR-2 localization at early stages (as proposed in their model figure). The authors should report whether the early cortical localization of AIR-1 requires microtubules by examining AIR-1 localization after nocodazole treatment.

2) The anterior PAR domain in air-1 embryos is accounted for based on curvature, and in this study there were no anterior PAR-2 domains in the control. However, the original PAR-2 GFP strains did show an anterior domain over the meiotic spindle (6/11 in. Cuenca et al.), and such a domain exists in mat arrested embryos, without depletion of air-1.

First, why is there a difference with the previous results – do the authors think it is because the original study used transgenic PAR-2 (i.e. excess PAR-2), while the current study uses a crispr'd GFP? But actually Cuenca cites that there is also an anterior PAR-2 domain in some embryos by Ab staining –.…is it a matter of detection? Regardless of the possible explanation, it would be nice to have this relationship addressed as it relates to potential mechanisms and the proposed model for AIR-1 moving from the cortex to the centrosome. For example, is there less cortical AIR-1 over the meiotic spindle region – i.e. is the spindle recruiting AIR-1 in a manner analogous to the manner proposed for centrosome? Likewise, is there less cortical AIR-1 at the posterior cortex as the centrosome matures in wild type? As part of addressing the comment #1, the authors would get the data (or have it already) on the relationship between this peripheral AIR-1 localization relative to the meiotic spindle and the centrosome in control embryos in this point #2. Also, the timing in general was not clearly explained. The model would predict that these symmetry breaking events in air-1 come up earlier than the normal domain- do they?

Revisions that must be addressed, but likely can be addressed via textual modifications:

1) The authors conclude that PAR-2 is recruited directly to curved regions. Is it possible that instead actomyosin flow is moving the PAR-2? It is noted that in some of the prior studies, the PAR-2 domain moves to the curved pole even if it starts laterally, and in one of the air-1(RNAi) videos here, the PAR-2 does seem to move. Was movement scored – was this common? This doesn't change the overall conclusions about curved regions and symmetry breaking –… but would affect parameters in the model.

2) The explanation for centrosome ablation experiments is somewhat confusing. We understand that brighter SAS7 allows earlier ablation, but why this would result in 2 PAR-2 domains that are centrosome independent, while Cowan's ablations resulted in no PAR-2 domain is not really explained (it doesn't "resolve the apparent discrepancy"). Can the authors provide more words in the Discussion section to help resolve the differences with the Cowan paper? For example, is it a matter of timing and Cowan did not look at the same time and so did not detect the domains? Was there something else different about the experiments? Can the new model presented account for the differences? This is important since a main conclusion of the paper – breaking of symmetry in the absence of centrosomes – is opposite to current dogma.

3) The microfabricated triangular shapes are very interesting. However, it seems odd that the authors chose to use triangles as the shape. We would think a diamond shape (with more acute angles at opposite ends, and more obtuse angles midway between the acute poles) would be more representative of the curvature in an embryo. Or perhaps a rectangle with right angles. Why did the authors choose the triangular shape? No need to be experimentally addressed, but the explanation on their choice would be appreciated.

4) The authors refer to PAR-2 preferring the angle with the highest degree of curvature, and yet the triangles appear to have the same angle at each corner. What do the authors mean by angles with more and less curvature in this context? It is not clear from the figure which angles are have the highest degree of curvature; they all look the same to me. It would seem to be interesting to apply the modeling to other shapes.

5) In Figure 3A, the schematics show centrioles in positions that do not at all correspond to the SAS-7 GFP signal. Do the schematics show centrioles from a different time point than the time point used to assess PAR-2 localization? This lack of correspondence between the schematics and the images on centriole position is confusing, and the authors need to explain this.

6) In Figure 5A, the authors initially use an endogenous GFP fusion to AIR-1 to show its cortical localization (presumably the authors used CRISPR to generate this endogenous fusion; this is not reported in the Materials and methods and the authors need to include in the Materials and methods how they obtained or made this endogenous fusion). However, in Figure 5C – 5F, the authors use an RNAi-resistant AIR-1 transgene (again not described in the Materials and methods as to how it was obtained or made, and this needs to be corrected) to assess AIR-1 and MT co-localization (and the dependence of cortical AIR-1 localization on spd-2 and spd-5). Why did the authors use an AIR-1 transgene and knockdown endogenous AIR-1 to assess this co-localization? Why not just use their endogenous fusion to AIR-1 to address the dependence of cortical AIR-1 on spd-2 and spd-5, and the co-localization of AIR-1 and MTs? The authors should explain why they use two different versions of AIR-1 in these experiments. Do the authors see the same dependence of cortical AIR-1 on spd-2 and spd-5 with the endogenous fusion? Why does cortical AIR-1 depend on SPD-2 and SPD-5? This finding is somewhat surprising (unexpected, as the authors put it), but it is not discussed in any detail. The authors should provide some discussion as to how SPD-2 and SPD-5 might be influencing cortical localization of AIR-1

7) In Figure 5I, how do the authors distinguish mCherry::PAR-2 at the cortex from GBP:mCherry::AIR-1 at the cortex (in the GFP::PH background)?

---

## [Author Response]

Essential revisions that require additional experiments:1) In Figure 5, the authors show that cortical AIR-1 co-localizes to a substantial extent with cortical microtubules (MTs), and that AIR-1 is not required for cortical MTs (if anything, cortical MTs are increased in the absence of AIR-1). However, the authors do not address whether cortical AIR-1 requires microtubules. They show earlier in the paper that the bipolarity phenotype observed after AIR-1 knockdown does not depend on MTs (even though it does depend on PAR-2 constructs that can bind MTs, and they address this paradox well in the Discussion). Furthermore, it is known that polarity establishment does not require MTs, as the authors point out. But given the authors' convincing documentation of AIR-1 and cortical microtubule co-localization, and the authors' use of nocodazole to examine the polarity phenotypes in the absence of microtubules and AIR-1, it is a very easy experiment to address the rather obvious question as to whether cortical AIR-1 requires microtubules: is AIR-1 still enriched at the cortex after strong nocodazole treatment? This is an important experiment: if AIR-1 does require microtubules to be at the cortex, then one would expect nocodazole treatment to phenocopy AIR-1 knockdown, and/or one would have to consider the possibility that AIR-1 does not need to be at the cortex to inhibit PAR-2 localization at early stages (as proposed in their model figure). The authors should report whether the early cortical localization of AIR-1 requires microtubules by examining AIR-1 localization after nocodazole treatment.

Thank you for suggesting this critical experiment. Accordingly, we treated embryos expressing endogenously tagged GFP::AIR-1 and mCherry::TBA-2 with nocodazole prior to cortical time-lapse microscopy. Importantly, we found that GFP::AIR-1 is absent from the cell cortex in such embryos, a result confirmed by immunostaining with AIR-1 antibodies in nocodazole-treated wild type embryos. These results demonstrate that microtubules are required for the cortical localization of AIR-1. These findings are reported in the new figure panels Figure 5E, I and Figure 5—figure supplement 6B, as well as in the corresponding Results section of the revised manuscript (subsection “Cortical AIR-1 prevents symmetry breaking early in the cell cycle”). Interestingly, in addition, we found that the kinase dead mutant version of AIR-1 does not localize to the cell cortex early in the cell cycle, in line with previous findings at later stages (Kotak et al., 2016). This new piece of data is reported in Figure 5J and also mentioned in the text of the revised manuscript (see the aforementioned subsection). Together, these new findings lead us to revise a tenet of our working model: we now propose that the role of AIR-1 in preventing aberrant polarity establishment early in the first cell cycle is exerted elsewhere than at the cell cortex, presumably in the cytoplasm. Given that the *Xenopus* homologue of SPD-2 is required for Aurora A activation (Joukov, Walter and De Nicolo, 2014), perhaps the lack of SPD-2 or SPD-5 in *C. elegans* likewise prevents AIR-1 activation, and thus, as a consequence, abolishes AIR-1 cortical distribution. These points are mentioned extensively in the Discussion of the revised manuscript (subsection “AIR-1 exerts a dual role in symmetry breaking”). In addition, the working model figure (now Figure 8) and accompanying text have been modified accordingly. Although these results establish that loss of cortical AIR-1 in otherwise wild type embryos does not affect A-P polarity in a significant manner, we also show that providing excess AIR-1 in the vicinity of the cortex is sufficient to prevent PAR-2 domain formation, perhaps by shifting the balance towards phosphorylation of a cortical substrate. That this might be the case is now discussed in the aforementioned subsection.

2) The anterior PAR domain in air-1 embryos is accounted for based on curvature, and in this study there were no anterior PAR-2 domains in the control. However, the original PAR-2 GFP strains did show an anterior domain over the meiotic spindle (6/11 in. Cuenca et al.), and such a domain exists in mat arrested embryos, without depletion of air-1.First, why is there a difference with the previous results – do the authors think it is because the original study used transgenic PAR-2 (i.e. excess PAR-2), while the current study uses a crispr'd GFP? But actually Cuenca cites that there is also an anterior PAR-2 domain in some embryos by Ab staining – is it a matter of detection? Regardless of the possible explanation, it would be nice to have this relationship addressed as it relates to potential mechanisms and the proposed model for AIR-1 moving from the cortex to the centrosome.

We never detected an anterior PAR-2 domain in the endogenously tagged GFP::PAR-2 strain used in this study (0/22 embryos analyzed). However, we found that GFP::PAR-2 indeed forms a small anterior domain when present in excess (13/16 embryos analyzed) (now reported in the new Figure 1—figure supplement 2A, B), as shown also by Cuenca et al. with another strain overexpressing PAR-2. The analysis of fixed specimens reported in the Boyd et al., 1996 paper referred to by Cuenca et al., 2003 shows that endogenous PAR-2 is present in a weak and uniform manner in some embryos during meiosis, and that the loss of this signal then occurs preferentially in the embryo center, resulting in transient weak anterior and posterior PAR-2 in some embryos early in the cell cycle. We repeated this analysis and found that wild-type embryos stained for endogenous PAR-2 only exceptionally exhibited a bipolar pattern early in the cell cycle (1/17 embryos analyzed), and never at pronuclear meeting (0/26 embryos analyzed) (Figure 1—figure supplement 2D, E). Importantly, otherwise stated, all our experiments relied on scoring endogenous GFP::PAR-2 or PAR-2 at pronuclear meeting, when no control embryo exhibited bipolarity (Figure 1—figure supplement 2C, D). Furthermore, scoring polarity phenotypes at the onset of pronuclear migration or at pronuclear meeting did not change the conclusions reached upon AIR-1-depletion (Figure 1—figure supplement 1C, D).

For example, is there less cortical AIR-1 over the meiotic spindle region – i.e. is the spindle recruiting AIR-1 in a manner analogous to the manner proposed for centrosome? Likewise, is there less cortical AIR-1 at the posterior cortex as the centrosome matures in wild type? As part of addressing the comment #1, the authors would get the data (or have it already) on the relationship between this peripheral AIR-1 localization relative to the meiotic spindle and the centrosome in control embryos in this point #2.

Prompted in part by the above comments, we have conducted spinning disc live imaging of embryos expressing GFP::AIR-1 in an attempt to address whether the cortical signal might be lower in the vicinity of the meiotic spindle early on, as well as in the vicinity of centrosomes at a later stage. Although our analysis did not reveal a difference in either location compared to the rest of the cell cortex, we think that further in depth imaging using other modalities with improved spatial and temporal resolution will be needed to ascertain this beyond doubt, and therefore chose not to report the findings to date in the present manuscript.

Also, the timing in general was not clearly explained. The model would predict that these symmetry breaking events in air-1 come up earlier than the normal domain- do they?

We apologize for not having been sufficiently clear on this important point. In fact, we found that PAR-2 domains appear approximately at the same time in *air-1(RNAi*) embryos as the single PAR-2 domain does in control embryos (as reported already in the initial submission). Therefore, AIR-1 is not required for delaying symmetry breaking in the wild type, where centrosomes are present. However, as shown in Figure 6D (previous Figure 5D), in conditions of delayed meiosis, where the centrosome-dependent polarization cue is not yet activated, the depletion of AIR-1 in the MAT-1 mutant leads to more robust formation of the anterior PAR-2 domain compared to the MAT-1 mutant alone. We have altered the text (Discussion) and the model figure (now Figure 8) accordingly in the revised manuscript.

Revisions that must be addressed, but likely can be addressed via textual modifications:1) The authors conclude that PAR-2 is recruited directly to curved regions. Is it possible that instead actomyosin flow is moving the PAR-2? It is noted that in some of the prior studies, the PAR-2 domain moves to the curved pole even if it starts laterally, and in one of the air-1(RNAi) videos here, the PAR-2 does seem to move. Was movement scored – was this common? This doesn't change the overall conclusions about curved regions and symmetry breaking – but would affect parameters in the model.

It appears that we have not been sufficiently clear on this point either. In control embryos, the PAR-2 domain is indeed relaxing towards the pole, likely by actomyosin flows, as already reported in Figure 4—figure supplement 1C of the original submission. We also observed this relaxation mechanism, at least to some extent, in *air-1(RNAi)* embryos. However, a complete lateral onset of polarity as in Figure 4—figure supplement 1C was rare in both control and *air-1(RNAi)* embryos. Nonetheless, we assessed the distance of the PAR-2 domain initiation site (or later the center of the established PAR domains) to the poles over time in a randomly chosen subset of control and *air-1(RNAi)* embryos (N=15 each) (new Figure 4—figure supplement 1D-F). These results demonstrate that control embryos initiate their posterior PAR-2 domain in close proximity to the pole and robustly correct this domain to exactly match the pole (Figure 4—figure supplement 1D). The offset of the PAR-2 initiation site from the pole did not differ in control and *air-1(RNAi)* embryos on the posterior side, whereas it arose with a larger offset on the anterior side, possibly because of invaginations resulting from the polar body extrusion that give rise to pronounced membrane curvatures (Figure 4—figure supplement 1E, F). Moreover, importantly, domain relaxation occurred for both anterior and posterior domains upon AIR-1 depletion, although in a less robust manner than in the wild type, likely due to impaired cortical flows (Figure 4—figure supplement 1D-F). In addition to being shown in the new Figure 4—figure supplement 1D-F, these points are mentioned in the subsection “Theoretical analysis of symmetry breaking in C. elegans zygotes”.

2) The explanation for centrosome ablation experiments is somewhat confusing. We understand that brighter SAS7 allows earlier ablation, but why this would result in 2 PAR-2 domains that are centrosome independent, while Cowan's ablations resulted in no PAR-2 domain is not really explained (it doesn't "resolve the apparent discrepancy"). Can the authors provide more words in the Discussion section to help resolve the differences with the Cowan paper? For example, is it a matter of timing and Cowan did not look at the same time and so did not detect the domains? Was there something else different about the experiments? Can the new model presented account for the differences? This is important since a main conclusion of the paper – breaking of symmetry in the absence of centrosomes – is opposite to current dogma.

Apologies here also for not having been sufficiently clear. In a nutshell, we think that the timing of the ablation might indeed be critical: since symmetry breaking as evidenced by cortical flows occurs before the posterior PAR-2 domain becomes detectable (as can be seen for instance in Figure 4—figure supplement 1C), it is plausible that centrosome ablation in the earlier study occurred when symmetry breaking had been initiated. Perhaps such initiation no longer allows for spontaneous symmetry breaking and instead results in the complete absence of cortical PAR-2. We have expanded the relevant Discussion paragraph to better explain our reasoning for the possible root of the apparent discrepancy between the two studies (subsection “Centrosome-independent polarization of the C. elegans zygote”).

3) The microfabricated triangular shapes are very interesting. However, it seems odd that the authors chose to use triangles as the shape. We would think a diamond shape (with more acute angles at opposite ends, and more obtuse angles midway between the acute poles) would be more representative of the curvature in an embryo. Or perhaps a rectangle with right angles. Why did the authors choose the triangular shape? No need to be experimentally addressed, but the explanation on their choice would be appreciated.

We chose the triangular shapes as we were curious to determine whether we could observe three PAR-2 domains in *air-1(RNAi*) embryos. Instead, AIR-1 depleted embryos established usually one and sometimes two PAR-2 domains in such triangular shapes, which turns out to be predicted by the physical model that was developed in parallel to the experimental work. Testing other shapes such as those proposed above will of course be interesting to consider in the future, also because this could help improve the model. That this will the case is now mentioned explicitly in the subsection “A physical basis for centrosome-independent polarization mechanisms”.

4) The authors refer to PAR-2 preferring the angle with the highest degree of curvature, and yet the triangles appear to have the same angle at each corner. What do the authors mean by angles with more and less curvature in this context? It is not clear from the figure which angles are have the highest degree of curvature; they all look the same to me. It would seem to be interesting to apply the modeling to other shapes.

In essence, embryos do not fill each corner of the triangular well to the same extent, thus leading to different degrees of curvature in the embryo proper. The metric was scored in all three corners prior to symmetry breaking, as was the localization of centrosomes and polar bodies. This analysis revealed that PAR-2 domains in *air-1(RNAi*) embryos usually localized in the corner in which the embryo exhibited the highest degree of curvature, as can be seen in Figure 3A and perhaps more clearly in the new Figure 3—figure supplement 1A. Furthermore, the text has been amended to clarify this point further (subsection “PAR-2 domains form preferentially in curved regions of the embryo”).

5) In Figure 3A, the schematics show centrioles in positions that do not at all correspond to the SAS-7 GFP signal. Do the schematics show centrioles from a different time point than the time point used to assess PAR-2 localization? This lack of correspondence between the schematics and the images on centriole position is confusing, and the authors need to explain this.

We apologize for the confusion. This is related to the above point: the schematic illustrated the position of centrosomes and polar bodies at the time of symmetry breaking, when their position was scored, whereas the microscopy data image showed the outcome of polarity at pronuclear meeting. We adjusted the schematic in Figure 3A to clarify this: we now show the localization of centrosomes before symmetry breaking, as previously, as well as at pronuclear meeting, which corresponds to the microscopy data images shown in the upper panel. Moreover, we have added a new Figure 3—figure supplement 1A showing the same embryos at the time the scoring was conducted.

6) In Figure 5A, the authors initially use an endogenous GFP fusion to AIR-1 to show its cortical localization (presumably the authors used CRISPR to generate this endogenous fusion; this is not reported in the Materials and methods and the authors need to include in the Materials and methods how they obtained or made this endogenous fusion). However, in Figure 5C – 5F, the authors use an RNAi-resistant AIR-1 transgene (again not described in the Materials and methods as to how it was obtained or made, and this needs to be corrected) to assess AIR-1 and MT co-localization (and the dependence of cortical AIR-1 localization on spd-2 and spd-5). Why did the authors use an AIR-1 transgene and knockdown endogenous AIR-1 to assess this co-localization? Why not just use their endogenous fusion to AIR-1 to address the dependence of cortical AIR-1 on spd-2 and spd-5, and the co-localization of AIR-1 and MTs? The authors should explain why they use two different versions of AIR-1 in these experiments. Do the authors see the same dependence of cortical AIR-1 on spd-2 and spd-5 with the endogenous fusion? Why does cortical AIR-1 depend on SPD-2 and SPD-5? This finding is somewhat surprising (unexpected, as the authors put it), but it is not discussed in any detail. The authors should provide some discussion as to how SPD-2 and SPD-5 might be influencing cortical localization of AIR-1

All *C. elegans* strains used in this work were referenced already in the initial submission in a list that can be found in the supplementary material, and which has been enhanced and updated. The reason for not having used the endogenously tagged GFP::AIR-1 strain to start with is that we obtained it at a later time than the strains expressing GFP::AIR-1 in addition to endogenous AIR-1. Importantly, however, both endogenously and exogenously tagged GFP::AIR-1 strains exhibited colocalization of GFP::AIR-1 with cortical microtubules early in the cell cycle as well as progressive recruitment to the centrosome. We repeated the experiment from Figure 5B in the endogenously tagged GFP::AIR-1 strain to replace the figure panel from the initial submission.

As for why AIR-1 might depend on SPD-2 and SPD-5, as mentioned also above in response to point 1, given that the *Xenopus* homologue of SPD-2 is required for Aurora A activation (Joukov et al., 2014), perhaps the lack of SPD-2 or SPD-5 in *C. elegans* likewise prevents AIR-1 activation, and thus, as a consequence, abolishes AIR-1 cortical distribution. These points are mentioned extensively in the Discussion of the revised manuscript (subsection “AIR-1 exerts a dual role in symmetry breaking”).

7) In Figure 5I, how do the authors distinguish mCherry::PAR-2 at the cortex from GBP:mCherry::AIR-1 at the cortex (in the GFP::PH background)?

The fluorescence intensity of membrane targeted GBP::mCherry::AIR-1 is much weaker that contributed by mCherry::PAR-2. Therefore, by using an exposure time adapted to visualize mCherry::PAR-2, the membrane bound GBP::mCherry::AIR-1 is barely visible and thus does not interfere with the scoring of the PAR-2 domain. That this is the case is now mentioned explicitly in Figure 5I figure legend.